# MILDLY CONSTRAINED EVALUATION POLICY FOR OFFLINE REINFORCEMENT LEARNING

## ABSTRACT

Offline reinforcement learning (RL) methodologies enforce constraints on the policy to adhere closely to the behavior policy, thereby stabilizing value learning and mitigating the selection of out-of-distribution (OOD) actions during test time. Conventional approaches apply identical constraints for both value learning and test time inference. However, our findings indicate that the constraints suitable for value estimation may in fact be excessively restrictive for action selection during test time. To address this issue, we propose a *Mildly Constrained Evaluation Policy (MCEP)* for test time inference with a more constrained *target policy* for value estimation. Since the *target policy* has been adopted in various prior approaches, MCEP can be seamlessly integrated with them as a plug-in. We instantiate MCEP based on TD3BC (Fujimoto & Gu, 2021), AWAC (Nair et al., 2020) and DQL (Wang et al., 2023) algorithms. The empirical results on D4RL MuJoCo locomotion and high-dimensional humanoid tasks show that the MCEP brought significant performance improvement on classic offline RL methods and can further improve SOTA methods. The codes are open-sourced at *link*.

## 1 INTRODUCTION

Offline reinforcement learning (RL) extracts a policy from data that is pre-collected by unknown policies. This setting does not require interactions with the environment thus it is well-suited for tasks where the interaction is costly or risky. Recently, it has been applied to Natural Language Processing (Snell et al., 2022), e-commerce (Degirmenci & Jones) and real-world robotics (Kalashnikov et al., 2021; Rafailov et al., 2021; Kumar et al., 2022; Shah et al., 2022) etc. Compared to the standard online setting where the policy gets improved via trial and error, learning with a static offline dataset raises novel challenges. One challenge is the distributional shift between the training data and the data encountered during deployment. To attain stable evaluation performance under the distributional shift, the policy is expected to stay close to the behavior policy. Another challenge is the "extrapolation error" (Fujimoto et al., 2019; Kumar et al., 2019) that indicates value estimate error on unseen state-action pairs or Out-Of-Distribution (OOD) actions. Worsely, this error can be amplified with bootstrapping and cause instability of the training, which is also known as deadly-triad (Van Hasselt et al., 2018). Majorities of model-free approaches tackle these challenges by either constraining the policy to adhere closely to the behavior policy (Wu et al., 2019; Kumar et al., 2019; Fujimoto & Gu, 2021) or regularising the Q to pessimistic estimation for OOD actions (Kumar et al., 2020; Lyu et al., 2022). In this work, we focus on *policy constraints* methods.

Policy constraints methods minimize the disparity between the policy distribution and the behavior distribution. It is found that policy constraints introduce a tradeoff between stabilizing value estimates and attaining better performance. While previous approaches focus on developing various constraints for the learning policy to address this tradeoff, the tradeoff itself is not well understood. Current solutions have confirmed that an excessively constrained policy enables stable value estimate but degrades the evaluation performance (Kumar et al., 2019; Singh et al., 2022; Yu et al., 2023). Nevertheless, it is not clear to what extent this constraint fails to stabilize value learning and to what extent this constraint leads to a performant evaluation policy. It is essential to investigate these questions as their answers indicate how well a solution can be found under the tradeoff. However, the investigation into the latter question is impeded by the existing tradeoff, as it requires tuning the constraint without influencing the value learning. To achieve this investigation, we circumvent the tradeoff and seek solutions for this investigation through the critic. For actor-critic

methods, (Czarnecki et al., 2019) has shed light on the potential of distilling a student policy that improves over the teacher using the teacher's critic. Inspired by this work, we propose to derive an extra *evaluation policy* from the critic. The *evaluation policy* does not join the policy evaluation step thus tunning its constraint does not influence value learning. The actor from the actor-critic is now called *target policy* as it is used only to stabilize the value estimation.

Based on the proposed framework, we empirically investigate the constraint strengths for 1) stabilizing value learning and 2) better evaluation performance. The results find that a milder constraint improves the evaluation performance but may fall beyond the constraint space of stable value estimation. This finding indicates that the optimal evaluation performance may not be found under the tradeoff, especially when stable value learning is the priority. Consequently, we propose a novel approach of using a *Mildly Constrained Evaluation Policy (MCEP)* derived from the critic to avoid solving the above-mentioned tradeoff and to achieve better evaluation performance.

As the *target policy* is commonly used in previous approaches, our MCEP can be integrated with them seamlessly. In this paper, we first validate the finding of (Czarnecki et al., 2019) in the offline setting by a toy maze experiment, where a constrained policy results in bad evaluation performance but its off-policy Q estimation indicates an optimal policy. After that, our experiments on D4RL (Fu et al., 2020) MoJoCo locomotion tasks showed that in most tasks, milder constraint achieves better evaluation performance while more restrictive constraint stabilizes the value estimate. Finally, we instantiated MCEP on TD3BC, AWAC and DQL algorithms. The empirical results of these instances on MuJoCo locomotion and high-dimensional humanoid tasks find that the MCEP brought significant performance improvement as it allows milder constraints without harming the value learning.

## 2 RELATED WORK

Policy constraints method (or behavior-regularized policy method) (Wu et al., 2019; Kumar et al., 2019; Siegel et al., 2020; Fujimoto & Gu, 2021) forces the policy distribution to stay close to the behavior distribution. Different discrepancy measurements such as KL divergence (Jaques et al., 2019; Wu et al., 2019), reverse KL divergence Cai et al. (2022) and Maximum Mean Discrepancy (Kumar et al., 2019) are applied in previous approaches. (Fujimoto & Gu, 2021) simply adds a behavior-cloning (BC) term to the online RL method Twin Delayed DDPG (TD3) (Fujimoto et al., 2018) and obtains competitive performances in the offline setting. While the above-mentioned methods calculate the divergence from the data, (Wu et al., 2022) estimates the density of the behavior distribution using VAE, and thus the divergence can be directly calculated. Except for explicit policy constraints, implicit constraints are achieved by different approaches. E.g. (Zhou et al., 2021) ensures the output actions stay in support of the data distribution by using a pre-trained conditional VAE (CVAE) decoder that maps latent actions to the behavior distribution. In all previous approaches, the constraints are applied to the learning policy that is queried during policy evaluation (value learning) and is evaluated in the environment during deployment. Our approach does not count on this learning policy for the deployment, instead, it is used as a *target policy* only for the value learning.

While it is well-known that a policy constraint can be efficient to reduce extrapolation errors, its drawback is not well-studied yet. (Kumar et al., 2019) reveals a tradeoff between reducing errors in the Q estimate and reducing the suboptimality bias that degrades the evaluation policy. A constraint is designed to create a policy space that ensures the resulting policy is under the support of the behavior distribution for mitigating bootstrapping error. (Singh et al., 2022) discussed the inefficiency of policy constraints on *heteroskedastic* dataset where the behavior varies across the state space in a highly non-uniform manner, as the constraint is state-agnostic. A reweighting method is proposed to achieve a state-aware distributional constraint to overcome this problem. Our work studies essential questions about the tradeoff (Kumar et al., 2019) and overcomes this overly restrictive constraint problem (Singh et al., 2022) by using an extra *evaluation policy*.

There are methods that extract an evaluation policy from a learned Q estimate. One-step RL (Brandfonbrener et al., 2021) first estimates the behavior policy and its Q estimate, which is later used for extracting the evaluation policy. Although its simplicity, one-step RL is found to perform badly in long-horizon problems due to a lack of iterative dynamic programming (Kostrikov et al., 2022). (Kostrikov et al., 2022) proposed Implicity Q learning (IQL) that avoids query of OOD actions by learning an upper expectile of the state value distribution. No explicit target policy is modeled during their Q learning. With the learned Q estimate, an evaluation policy is extracted using

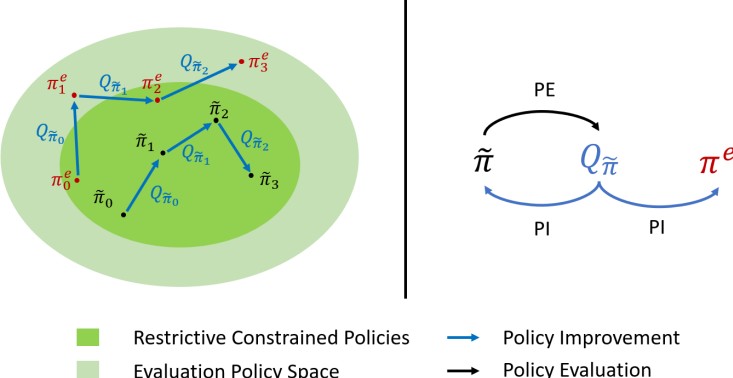

Figure 1: **Left:** diagram depicts policy trajectories for target policy $\tilde{\pi}$ and MCEP $\pi^e$. **Right:** policy evaluation steps to update $Q_{\tilde{\pi}}$ and policy improvement steps to update $\tilde{\pi}$ and $\pi^e$.

advantage-weighted regression (Wang et al., 2018; Peng et al., 2019). Our approach has a similar form of extracting an evaluation policy from a learned Q estimate. However, one-step RL aims to avoid distribution shift and iterative error exploitation during iterative dynamic programming. IQL avoids error exploitation by eliminating OOD action queries and abandoning policy improvement (i.e. the policy is not trained against the Q estimate). Our work instead tries to address the error exploitation problem and evaluation performance by using policies of different constraint strengths.

## 3 BACKGROUND

We model the environment as a Markov Decision Process (MDP) $\langle S, A, R, T, p_0(s), \gamma, \rangle$, where $S$ is the state space, $A$ is the action space, $R$ is the reward function, $T(s'|s, a)$ is the transition probability, $p_0(s)$ is initial state distribution and $\gamma$ is a discount factor. In the offline setting, a static dataset $\mathcal{D}_\beta = \{(s, a, r, s')\}$ is pre-collected by a behavior policy $\pi_\beta$. The goal is to learn a policy $\pi_\phi(s)$ with the dataset $\mathcal{D}$ that maximizes the discounted cumulated rewards in the MDP:

$$\phi^* = \arg\max_\phi \mathbb{E}_{s_0 \sim p_0(\cdot), a_t \sim \pi_\phi(s_t), s_{t+1} \sim T(\cdot|s_t, a_t)} [\sum_{t=0}^{\infty} \gamma^t R(s_t, a_t)] \tag{1}$$

Next, we introduce the general policy constraint method, where the policy $\pi_\phi$ and an off-policy Q estimate $Q_\theta$ are updated by iteratively taking policy improvement steps and policy evaluation steps, respectively. The policy evaluation step minimizes the Bellman error:

$$\mathcal{L}_Q(\theta) = \mathbb{E}_{s_t, a_t \sim \mathcal{D}, a_{t+1} \sim \pi_\phi(s_{t+1})} [(Q_\theta(s_t, a_t) - (r + \gamma Q_{\theta'}(s_t, a_{t+1})))^2]. \tag{2}$$

where the $\theta'$ is the parameter for a delayed-updated target Q network. The Q value for the next state is calculated with actions $a_{t+1}$ from the learning policy that is updated through the policy improvement step:

$$\mathcal{L}_\pi(\phi) = \mathbb{E}_{s \sim \mathcal{D}, a \sim \pi_\phi(s)} [-Q_\theta(s, a) + wC(\pi_\beta, \pi_\phi)], \tag{3}$$

where $C$ is a constraint measuring the discrepancy between the policy distribution $\pi_\phi$ and the behavior distribution $\pi_\beta$. The $w \in (0, \infty]$ is a weighting factor. Different kinds of constraints were used such as Maximum Mean Discrepancy (MMD), KL divergence, and reverse KL divergence.

## 4 METHOD

In this section, we first introduce the generic algorithm that can be integrated with any policy constraints method. Next, we introduce three examples based on offline RL methods TD3BC, AWAC and DQL. With a mildly constrained evaluation policy, we name these three instances as *TD3BC-MCEP, AWAC-MCEP* and *DQL-MCEP*.

## 4.1 OFFLINE RL WITH MILDLY CONSTRAINED EVALUATION POLICY

The proposed method is designed to overcome the tradeoff between stable value learning and a performant evaluation policy. In previous constrained policy methods, a restrictive policy constraint is applied to obtain stable value learning. We retain this benefit but use this policy (actor) $\tilde{\pi}_\psi$ as a *target policy* only to obtain stable value learning. To achieve better evaluation performance, we introduce an MCEP $\pi_\phi^e$ that is updated by taking policy improvement steps with the critic $Q_{\tilde{\pi}_\psi}$. Different from $\tilde{\pi}_\psi$, $\pi_\phi^e$ does not participate in the policy evaluation procedure. Therefore, a mild policy constraint can be applied, which helps $\pi_\phi^e$ go further away from the behavior distribution without influencing the stability of value learning. We demonstrate the policy spaces and policy trajectories for $\tilde{\pi}_\psi$ and $\pi_\phi^e$ in the l.h.s. diagram of Figure 1, where $\pi_\phi^e$ is updated in the wider policy space using $Q_{\tilde{\pi}_\psi}$.

The overall algorithm is shown as pseudo-codes (Alg. 1). At each step, the $Q_{\tilde{\pi}_\psi}$, $\tilde{\pi}_\psi$ and $\pi_\phi^e$ are updated iteratively. A policy evaluation step updates $Q_{\tilde{\pi}_\psi}$ by minimizing the TD error (line 4), i.e. the deviation between the approximate $Q$ and its target value. Next, a policy improvement step updates $\tilde{\pi}_\psi$ (line 6. These two steps form the actor-critic algorithm. After that, $\pi_\phi^e$ is extracted from the $Q_{\tilde{\pi}_\psi}$, by taking a policy improvement step with a policy constraint that is likely milder than the constraint for $\tilde{\pi}_\psi$ (line 7). Many approaches can be taken to obtain a milder policy constraint. For example, tuning down the weight factor $w^e$ for the policy constraint term or replacing the constraint measurement with a less restrictive one. Note that

---

**Algorithm 1** MCEP Training

1: **Hyperparameters:** LR $\alpha$, EMA $\eta$, $\tilde{w}$ and $w^e$

2: **Initialize:** $\theta, \theta', \psi$, and $\phi$
3: **for** i=1, 2, ..., N **do**
4:     $\theta \leftarrow \theta - \alpha \mathcal{L}_Q(\theta)$ (Equation 2)
5:     $\theta' \leftarrow (1 - \eta)\theta' + \eta\theta$
6:     $\psi \leftarrow \psi - \alpha \mathcal{L}_{\tilde{\pi}}(\psi; \tilde{w})$ (Equation 3)
7:     $\phi \leftarrow \phi - \alpha \mathcal{L}_{\pi^e}(\phi; w^e)$ (Equation 3)

---

the constraint for $\pi_\phi^e$ is necessary (the constraint term should not be dropped) as the $Q_{\tilde{\pi}_\psi}$ has large approximate errors for state-action pairs that are far from the data distribution.

As the evaluation policy $\pi_\phi^e$ is not involved in the actor-critic updates, one might want to update $\pi_\phi^e$ after the convergence of the $Q_{\tilde{\pi}_\psi}$. An experiment to compare these design options can be found in the Appendix Section A.5. Algorithm 1 that simultaneously updates two policies and these updates (line 6 and 7) can be parallelized to achieve little extra training time based on the base algorithm.

## 4.2 THREE EXAMPLES: TD3BC-MCEP, AWAC-MCEP AND DQL-MCEP

**TD3BC with MCEP** TD3BC takes a minimalist modification on the online RL algorithm TD3. To keep the learned policy to stay close to the behavior distribution, a behavior-cloning term is added to the policy improvement objective. TD3 learns a deterministic policy therefore the behavior cloning is achieved by directly regressing the data actions. For TD3BC-MCEP, the *target policy* $\tilde{\pi}_\psi$ has the same policy improvement objective as TD3BC:

$$\mathcal{L}_{\tilde{\pi}}(\psi) = \mathbb{E}_{(s,a)\sim\mathcal{D}}[-\tilde{\lambda}Q_\theta(s, \tilde{\pi}_\psi(s)) + (a - \tilde{\pi}_\psi(s))^2], \tag{4}$$

where the $\tilde{\lambda} = \frac{\tilde{\alpha}}{\frac{1}{N}\sum_{s_i,a_i}|Q_\theta(s_i,a_i)|}$ is a normalizer for Q values with a hyper-parameter $\tilde{\alpha}$: The $Q_\theta$ is updated with the policy evaluation step similar to Eq. 2 using $\tilde{\pi}_\psi$. The MCEP $\pi_\phi^e$ is updated by policy improvement steps with the $Q_{\tilde{\pi}}$ taking part in. The policy improvement objective function for $\pi_\phi^e$ is similar to Eq. 4 but with a higher-value $\alpha^e$ for the Q-value normalizer $\lambda^e$. The final objective for $\pi_\phi^e$ is

$$\mathcal{L}_{\pi^e}(\phi) = \mathbb{E}_{(s,a)\sim\mathcal{D}}[-\lambda^e Q(s, \pi_\phi^e(s)) + (a - \pi_\phi^e(s))^2]. \tag{5}$$

**AWAC with MCEP** AWAC (Nair et al., 2020) is an advantage-weighted behavior cloning method. As the target policy imitates the actions from the behavior distribution, it stays close to the behavior distribution during learning. In AWAC-MCEP, the policy evaluation follows the Eq. 2 with the target policy $\tilde{\pi}_\psi$ that updates with the following objective:

$$\mathcal{L}_{\tilde{\pi}}(\psi) = \mathbb{E}_{(s,a)\sim\mathcal{D}}\left[-\exp\left(\frac{1}{\tilde{\lambda}}A(s,a)\right)\log\tilde{\pi}_\psi(a|s)\right], \tag{6}$$

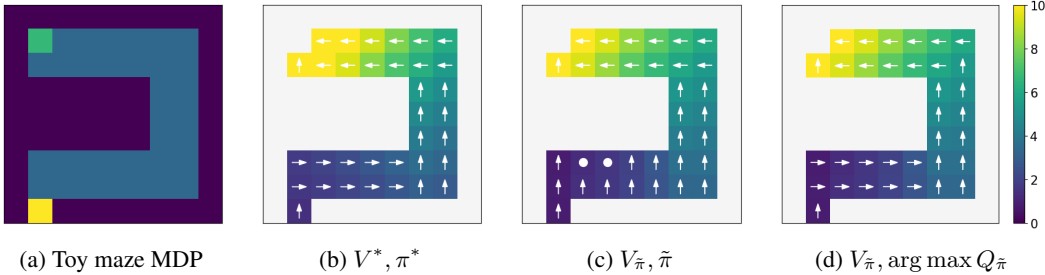

|(a) Toy maze MDP | (b) $V^*, \pi^*$ | (c) $V_{\tilde{\pi}}, \tilde{\pi}$ | (d) $V_{\tilde{\pi}}, \arg\max Q_{\tilde{\pi}}$ |

Figure 2: Evaluation of policy constraint method on a toy maze MDP 2a. In other figures, the color of a grid represents the state value and arrows indicate the actions from the corresponding policy. 2b shows the optimal value function and one optimal policy. 2c shows a constrained policy trained from the above-mentioned offline data, with its value function calculated by $V_\pi = \mathbb{E}_a Q(s, \pi(a|s))$. The policy does not perform well in the low state-value area but its value function is close to the optimal value function. 2d indicates that an optimal policy is recovered by deriving the greedy policy from the off-policy Q estimate (the critic).

where the advantage $A(s,a) = Q_\theta(s,a) - Q_\theta(s, \tilde{\pi}_\psi(s))$. This objective function solves an advantage-weighted maximum likelihood. Note that the gradient will not be passed through the advantage term. As this objective has no policy improvement term, we use the original policy improvement with KL divergence as the policy constraint and construct the following policy improvement objective:

$$\mathcal{L}_{\pi^e}(\phi) = \mathbb{E}_{s,a\sim\mathcal{D},\hat{a}\sim\pi^e(\cdot|s)}[-A(s,\hat{a}) + \lambda^e D_{KL}(\pi_\beta(\cdot|s)\|\pi^e_\phi(\cdot|s))] \tag{7}$$

$$= \mathbb{E}_{s,a\sim\mathcal{D},\hat{a}\sim\pi^e(\cdot|s)}[-A(s,\hat{a}) - \lambda^e \log\pi^e_\phi(a|s)], \tag{8}$$

where the weighting factor $\lambda^e$ is a hyper-parameter. Although the Eq. 6 is derived by solving Eq. 8 in a parametric-policy space, the original problem (Eq. 8) is less restrictive even with $\tilde{\lambda} = \lambda^e$ as the gradient back-propagates through the $-A(s, \pi^e(s))$ term. This difference means that even with a $\lambda^e > \tilde{\lambda}$, the policy constraint for $\pi^e$ could still be more relaxed than the policy constraint for $\tilde{\pi}$.

**DQL with MCEP** Diffusion Q-Learning (Wang et al., 2023) is one of the SOTA offline RL methods that applied a highly expressive conditional diffusion model as the policy to handle multimodal behavior distribution. Its policy improvement step is

$$\mathcal{L}_{\tilde{\pi}}(\psi) = \mathbb{E}_{s\sim\mathcal{D},a\sim\tilde{\pi}}[-\tilde{\lambda}Q(s,a) + C(\pi_\beta, \tilde{\pi})], \tag{9}$$

where $C(\pi_\beta, \tilde{\pi})$ is a behavior cloning term and $\tilde{\lambda}$ is the Q normalizer, similar to TD3BC. ~~We next introduce the policy improvement step for the evaluation policy.~~ The policy improvement step for the evaluation policy has the same manner as the target policy, except for using a different constraint strength.

$$\mathcal{L}_{\pi^E}(\phi) = \mathbb{E}_{s\sim\mathcal{D},a\sim\pi^E}[-\lambda^E Q(s,a) + C(\pi_\beta, \pi^E)]. \tag{10}$$

## 5 EXPERIMENTS

In this section, we set up 4 groups of experiments to illustrate: 1) the policy constraint might degrade the evaluation performance by forcing the policy to stay close to low-state-value transitions. 2) Milder policy constraints might achieve performance improvement but also make unstable Q estimate. 3) The evaluation policy allows milder policy constraints without influencing Q estimate 4) Our method brought significant performance improvement compared to the target policy on MuJoCo locomotion and high-dimensional humanoid tasks. ~~4)~~ 5) the MCEP generally gains a higher estimated Q compared to the target policy. Additionally, we adopt 2 groups of ablation studies to verify the benefit of an MCEP and to investigate the constraint strengths of MCEP.

**Environments** D4RL (Fu et al., 2020) is an offline RL benchmark consisting of many task sets. Our experiments select 3 versions of MuJoCo locomotion (-v2) datasets: data collected by rolling out a

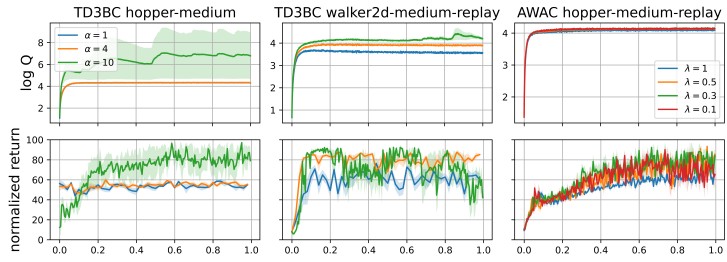
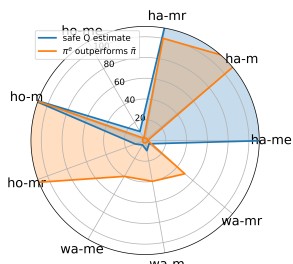

Figure 4: The training process of TD3BC and AWAC. **Left:** TD3BC on *hopper-medium-v2*. **Middle:** TD3BC on *walker2d-medium-replay-v2*. **Right:** AWAC on *hopper-medium-replay-v2*.

Figure 5: $\alpha$ values in TD3BC for value estimate and test time inference in MuJoCo locomotion tasks.

medium-performance policy (*medium*), the replay buffer during training a medium-performance policy (*medium-replay*), a $50\% - 50\%$ mixture of the medium data and expert demonstrations (*medium-expert*). To investigate more challenging high-dimensional tasks, we additionally collect 3 datasets for Humanoid-v2 tasks following the same collecting approach of D4RL: *humanoid-medium-v2, humanoid-medium-replay-v2, humanoid-medium-expert-v2*. The humanoid-v2 task has an observation space of 376 dimension and an action space of 17 dimension. This task is not widely used in offline RL research. (Wang et al., 2020; Bhargava et al., 2023) considers this task but our data is independent of theirs. Compared to (Bhargava et al., 2023), we do not consider pure expert data but include the *medium-replay* to study the replay buffer. The statistics of humanoid datasets are listed in Table 4.

## 5.1 TARGET POLICY THAT ENABLES SAFE Q ESTIMATE MIGHT BE OVERLY CONSTRAINED

To investigate the policy constraint under a highly suboptimal dataset, we set up a toy maze MDP that is similar to the one used in (Kostrikov et al., 2022). The environment is depicted in Figure 2a, where the lower left yellow grid is the starting point and the upper left green grid is the terminal state that gives a reward of 10. Other grids give no reward. Dark blue indicates un-walkable areas. The action space is defined as 4 direction movements (arrows) and staying where the agent is (filled circles). There is a $25\%$ probability that a random action is taken instead of the action from the agent. For the dataset, 99 trajectories are collected by a uniformly random agent and 1 trajectory is collected by an expert policy. Fig. 2b shows the optimal value function (colors) and one of the optimal policies.

We trained a constrained policy using Eq. 2 and Eq. 8 in an actor-critic manner, where the actor is constrained by a KL divergence with a weight factor of 1. Figure 2c shows the value function and the policy. We observe that the learned value function is close to the optimal one in Figure 2b. However, the policy does not make optimal actions in the lower left areas where the state values are relatively low. As the policy improvement objective shows a trade-off between the Q and the KL divergence, when the Q value is low, the KL divergence term will obtain higher priority. In other words, in low-Q-value areas, the KL divergence takes the majority for the learning objective, which makes the policy stay closer to the transitions in low-value areas. However, we find that the corresponding value function indicates an optimal policy. In Figure 2d, we recover a greedy policy underlying the learned critic that shows an optimal policy. This approach of utilizing the value function of the imperfect teacher policy is originally suggested by Czarnecki et al. (2019). In conclusion, the constraint might degrade the evaluation performance although the learned critic may indicate a better policy. Although such a trade-off between the Q term and the KL divergence term can be alleviated in previous work (Fujimoto & Gu, 2021) by normalizing the Q values, in the next section, we will illustrate that the constraint required to obtain performant evaluation policy can still cause unstable value estimate.

| Task Name | BC | 10%-BC | CQL | IQL | TD3BC | TD3BC-MCEP (ours) | AWAC | AWAC-MCEP (ours) | EQL | DQL | DQL-MCEP (ours) |
|---|---|---|---|---|---|---|---|---|---|---|---|
| halfcheetah-m | 42.4±0.1 | 43.1±0.3 | 44.0 | 47.4±0.1 | 48.7±0.2 | **55.5±0.4** | 45.1±0 | 46.9±0 | 46.5±0.1 | 49.8±0.2 | 53.2±0.2 |
| hopper-m | 54.1±1.1 | 56.9±1.6 | 58.5 | 65±3.6 | 56.1±1.2 | 91.8±0.9 | 58.9±1.9 | **98.1±0.6** | 67±1.3 | 81.7±6.6 | 95.5±2.2 |
| walker2d-m | 71±1.7 | 73.3±2.5 | 72.5 | 80.4±1.7 | 85.2±0.9 | **88.8±0.5** | 79.6±1.5 | 81.4±1.6 | 81.8±1.1 | 85.5 ± 0.8 | 75.3±3.6 |
| halfcheetah-m-r | 37.8±1.1 | 39.9±0.8 | 45.5 | 43.2±0.8 | 44.8±0.3 | **50.6±0.2** | 43.3±0.1 | 44.9±0.1 | 43.1±0.5 | 47±0.2 | 47.8±0.1 |
| hopper-m-r | 22.5±3.0 | 72±2.1 | 95.0 | 74.2±5.3 | 55.2±10.8 | 100.9±0.4 | 64.8±6.2 | **101.1±0.2** | 87.9±19.1 | 100.6±0.2 | 100.9±0.3 |
| walker2d-m-r | 14.4±2.7 | 56.6±3.3 | 77.2 | 62.7±1.9 | 50.9±16.1 | 86.3±3.2 | 84.1±0.6 | 83.4±0.8 | 71.4±4.7 | **93.6±2.5** | 92.6±2.1 |
| halfcheetah-m-e | 62.3±1.5 | 93.5±0 | 91.6 | 91.2±1.0 | 87.1±1.4 | 71.5±3.7 | 77.6±2.6 | 69.5±3.8 | 89.4±1.6 | **95.7±0.4** | 93.4±0.8 |
| hopper-m-e | 52.5±1.4 | 108.9±0.0 | 105.4 | **110.2±0.3** | 91.7±10.5 | 80.1±12.7 | 52.4±8.7 | 84.3±16.4 | 97.3±3.3 | 102.1±3.0 | 107.7 ± 1.5 |
| walker2d-m-e | 107±1.1 | 111.1±0.5 | 108.8 | **111.1±0.5** | 110.4±0.5 | **111.7±0.3** | 109.5±0.2 | 110.1±0.2 | 109.8±0.0 | 109.5±0.1 | 109.7±0.0 |
| Average | 51.5 | 72.8 | 77.6 | 76.1 | 70.0 | 81.9 | 68.3 | 79.9 | 77.1 | 85 | **86.2** |

Table 1: Normalized episode returns on D4RL benchmark. The results (except for CQL) are means and standard errors from the last step of 5 runs using different random seeds. Performances that are higher than corresponding baselines are underlined and task-wise best performances are bolded.

## 5.2 EVALUATION POLICY ALLOWS MILDER CONSTRAINTS

The previous experiment shows that a restrictive constraint might harm the test-time inference, which motivates us to investigate milder policy constraints. Firstly, we relax the policy constraint on TD3BC and AWAC by setting up different hyper-parameter values that control the strengths of the policy constraints. For TD3BC, we set $\alpha = \{1, 4, 10\}$ ((Fujimoto & Gu, 2021) recommends $\alpha = 2.5$). For AWAC, we set $\lambda = \{1.0, 0.5, 0.3, 0.1\}$ ((Nair et al., 2020) recommends $\lambda = 1$). Finally, We visualize the evaluation performance and the learned Q estimates.

In Figure 4, the left two columns show the training of TD3BC in the *hopper-medium-v2* and *walker2d-medium-replay-v2*. In both domains, we found that using a milder constraint by tuning the $\alpha$ from 1 to 4 improves the evaluation performance, which motivates us to expect better performance with $\alpha = 10$. See from the normalized return of $\alpha = 10$, we do observe higher performances. However, the training is unstable because of the divergence in value estimate and thus the policy performance is also unsteady. This experiment indicates the tradeoff between the stable Q estimate and the evaluation performance. The rightmost column shows the training of AWAC in *hopper-medium-replay-v2*, we observe higher evaluation performance by relaxing the constraint ($\lambda > 1$). Although the Q estimate keeps stable during the training in all $\lambda$ values, higher $\lambda$ still result in unstable policy performance and causes the performance crash with $\lambda = 0.1$.

Concluding on all these examples, a milder constraint can potentially improve the performance but may cause unstable Q estimates or unstable policy performances. As we find that relaxing the constraint on current methods triggers unstable training, which hinders the investigation of milder constraints on their policy performance. We instead systematically study the constraint strengths in TD3BC and TD3BC with *evaluation policy* (TD3BC-EP).

We first tune the $\alpha$ for TD3BC to unveil the range for safe Q estimates. Then in TD3BC-EP, we tune the $\alpha^e$ for the evaluation policy with a fixed $\tilde{\alpha} = 2.5$ to see the policy performance under a stable Q estimate. The $\alpha$ ($\alpha^e$) is tuned within $\{2.5, 5, 10, 20, 30, 40, 50, 60, 70, 80, 90, 100\}$. For each $\alpha$ ($\alpha^e$), we deploy 5 training with different random seeds. In Figure 5, we visualize two constraint ranges for MuJoCo locomotion tasks. The blue area shows $\alpha$ values where the constraint strength enables a stable Q estimate for all seeds. The edge of blue area shows the lowest $\alpha$ value that causes Q value explosion. The orange area shows the range of $\alpha^e$ where the learned evaluation policy outperforms the target policy. Its edge (the orange line) shows the lowest $\alpha^e$ values where the evaluation policy performance is worse than the target policy. For each task, the orange area has a lower bound $\alpha^e = 2.5$ where the evaluation policy shows a similar performance to the target policy.

Note that $\alpha$ weighs the Q term and thus a larger $\alpha$ indicates a less restrictive constraint. Comparing the blue area and the orange area, we observe that in 7 out of the 9 tasks (7 axis where the orange range is not zero), the evaluation policy achieves better performance than the target policy. In 5 tasks (5 axis where the orange range is larger than the blue one), the evaluation policy allows milder policy constraints which cause unsafe q estimate in TD3BC. In conclusion, evaluation policy allows milder policy constraints for potentially better performance and does not influence the Q estimate.

## 5.3 COMPARISON ON MUJOCO LOCOMOTION TASKS

**D4RL MuJoCo Locomotion** We compare the proposed method to behavior cloning, classic offline RL baselines AWAC, TD3BC, CQL and IQL, along with SOTA offline RL methods Extreme Q-Learning (EQL) (Garg et al., 2023) and DQL. For D4RL datasets, similar hyperparameters are used. The baseline methods (TD3BC, AWAC and DQL) use the hyper-parameter recommended by their papers. TD3BC uses $\alpha = 2.5$ for its Q value normalizer, AWAC uses $1.0$ for the advantage value normalizer and DQL uses $\alpha = 1.0$. In TD3BC-MCEP, the target policy uses $\tilde{\alpha} = 2.5$ and the MCEP uses $\alpha^e = 10$. In AWAC-MCEP, the target policy has $\tilde{\lambda} = 1.0$ and the MCEP has $\lambda^e = 0.6$. In DQL-MCEP, $\tilde{\alpha} = 1.0$ for target policy and $\alpha^e = 2.5$ for evaluation policy. The full list of hyper-parameters can be found in Table 2.

As is shown in Table 1, we observe that the evaluation policies with a mild constraint significantly outperform their corresponding target policy. TD3BC-MCEP gains progress on all *medium* and *medium-replay* datasets. Although the progress is superior, we observe a performance degradation on the *medium-expert* datasets which indicates an overly relaxed constraint for the evaluation policy. Nevertheless, the TD3BC-MCEP achieves much better general performance than the target policy. In the AWAC-MCEP, we observe a consistent performance improvement over the target policy on most tasks. Additionally, evaluation policies from both TD3BC-MCEP and AWAC-MCEP outperform the CQL, IQL and EQL while the target policies have relatively low performances. On the SOTA method, DQL, the MCEP can still obtain further performance improvement.

**Humanoid** One of the major challenges for offline RL is the distributional shift. In high-dimensional environments, this challenge is exaggerated as the collected data is relatively limited. To evaluate the proposed method on the ability to handle these environments, we compare the TD3BC-MCEP with IQL, CRR, TD3BC, BC and be-

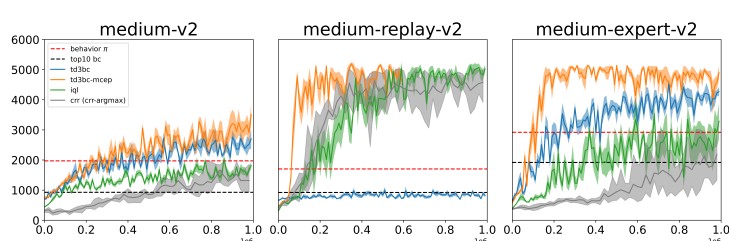

Figure 6: The visualization of the training on three humanoid tasks.

havior policy. As seen in Figure 6, TD3BC-MCEP achieves higher returns in *medium* and *medium-expert*, both are collected by rolling out the learned online policy. In *medium-replay*, the replay buffer of the online training, TD3BC-MCEP also achieves superior performance and shows a faster convergence rate than IQL. See Section A.2 for more details about this experiments.

## 5.4 ABLATION STUDY

In this section, we design 2 groups of ablation studies to investigate the effect of the extra evaluation policy and its constraint strengths. Reported results are averaged on 5 random seeds.

**Performance of the extra evaluation policy.** Now, we investigate the performance of the introduced evaluation policy $\pi^e$. For TD3BC, we set the parameter $\alpha = \{2.5, 10.0\}$. A large $\alpha$ indicates a milder constraint. After that, we train TD3BC-MCEP with $\tilde{\alpha} = 2.5$ and $\alpha^e = 10.0$. For AWAC, we trained AWAC with the $\lambda = \{1.0, 0.5\}$ and AWAC-MCEP with $\tilde{\lambda} = 1.0$ and $\lambda^e = 0.5$.

The results are shown in Figure 7. The scores for different datasets are grouped for each domain. By com-

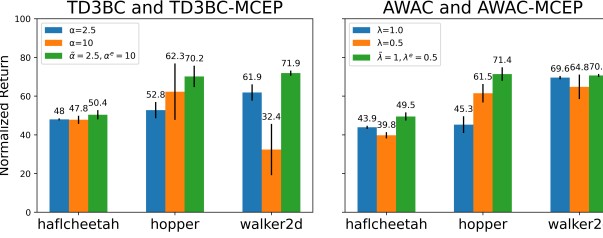

Figure 7: **Left**: TD3BC with $\alpha = 2.5$, $\alpha = 10$ and TD3BC-MCEP with $\tilde{\alpha} = 2.5$, $\alpha^e = 10$. **Right:** AWAC with $\lambda = 1.0$, $\lambda = 0.5$ and AWAC-MCEP with $\tilde{\lambda} = 1.0$ and $\lambda^e = 0.5$.

paring TD3BC of different $\alpha$ values, we found a milder constraint ($\alpha = 10.0$) brought performance improvement in hopper tasks but degrades the performance in walker2d tasks. The degradation is potentially caused by unstable value estimates (see experiment at section 5.2). Finally, the *evaluation policy* ($\alpha^E = 10.0$) with a *target policy* of $\tilde{\alpha} = 2.5$ achieves the best performance in all three tasks. In AWAC, a lower $\lambda$ value brought policy improvement in hopper tasks but degrades performances in half-cheetah and walker2d tasks. Finally, an evaluation policy obtains the best performances in all tasks.

In conclusion, we observe consistent performance improvement brought by an extra MCEP that circumvents the tradeoff brought by the constraint.

**Constraint strengths of the evaluation policy.** We set up two groups of ablation experiments to investigate the evaluation policy performance under different constraint strengths. For TD3BC-MCEP, we tune the constraint strength by setting the Q normalizer hyper-parameter $\alpha$. The target policy is fixed to $\tilde{\alpha} = 2.5$. We pick three strengths for evaluation policy $\alpha^e = \{1.0, 2.5, 10.0\}$ to create more restrictive, similar, and milder constraints, respectively. For AWAC-MCEP, the target policy has $\tilde{\lambda} = 1.0$. However, it is not straightforward to create a similar constraint for

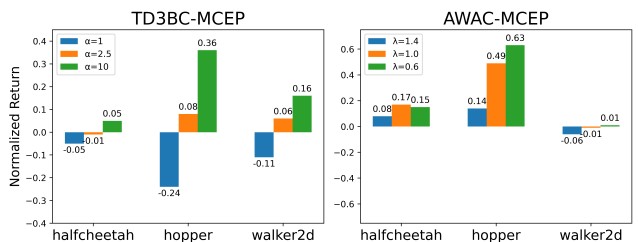

Figure 8: **Left:** TD3BC-EP with $\alpha = 1.0$, $\alpha = 2.5$ and $\alpha = 10.0$. **Right:** AWAC-EP with $\lambda = 1.4$, $\lambda = 1.0$ and $\lambda = 0.6$.

the evaluation policy as it has a different policy improvement objective. We set $\lambda^e = \{0.6, 1.0, 1.4\}$ to show how performance changes with different constraint strengths.

The performance improvements over the target policy are shown in Figure 8. For TD3BC-MCEP, a more restrictive constraint ($\alpha^e = 1.0$) for the evaluation causes a significant performance drop. With a similar constraint ($\tilde{\alpha} = \alpha^e = 2.5$), the performance is slightly improved in two domains. When the evaluation policy has a milder constraint ($\alpha^e = 10$), significant performance improvements are observed in all 3 domains. The right column presents the results of AWAC-MCEP. Generally, the performance in hopper tasks keeps increasing with milder constraints (smaller $\lambda$) while the half-cheetah and walker2d tasks show performances that are enhanced from $\lambda = 1.4$ to $\lambda = 1$ and similar performances between $\lambda = 1$ and $\lambda = 0.6$. It is worth noting that the evaluation policy consistently outperforms the target policy in halfcheetah and hopper domains. On the walker2d task, a strong constraint ($\lambda = 1.4$) causes a performance degradation.

In conclusion, for both algorithms, we observe that on evaluation policy, a milder constraint obtains higher performance than the target policy while a restrictive constraint may harm the performance.

### 5.5 ESTIMATED Q VALUES FOR THE LEARNED EVALUATION POLICIES

To compare the performance of the policies on the learning objective (maximizing the Q values), we visualze Q differences between the policy action and the data action $Q(s, \pi(s)) - Q(s, a)$ in the training data (Figure 12, 13). We find that both the target policy and the MCEP have larger Q estimations than the behavior actions. Additionally, MCEP generally has higher Q values than the target policy, indicating that the MCEP is able to move further toward large Q values.

## 6 CONCLUSION

This work focuses on the policy constraints methods where the constraint addresses the tradeoff between stable value estimate and evaluation performance. While to what extent the constraint achieves the best results for each end of this tradeoff remains unknown, we first investigate the constraint strength range for a stable value estimate and for evaluation performance. Our findings indicate that test time inference requires milder constraints that can go beyond the range of stable

value estimates. We propose to use an auxiliary *mildly constrained evaluation policy* to circumvent the above-mentioned tradeoff and derive a performant evaluation policy. The empirical results on 3 policy constraints methods show that MCEP is general and can obtain significant performance improvement. The evaluation on high-dimensional humanoid tasks verifies that the proposed method is powerful to tackle distributional shifts.

**Limitations.** Although the MCEP is able to obtain a better performance, it depends on stable value estimation. Unstable value learning may crash both the target policy and the evaluation policy. While the target policy may recover its performance by iterative policy improvement and policy evaluation, we observe that the evaluation policy may fail to do so. Therefore, a restrictive constrained target policy that stabilizes the value learning is essential for the proposed method.

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

# A APPENDIX

## A.1 IMPLEMENTATIONS AND HYPER-PARAMETERS FOR D4RL TASKS

For CQL, we reported the results from the IQL paper (Kostrikov et al., 2022) to show CQL results on "-v2" tasks. For IQL, we use the official implementation (Kostrikov, 2022) to obtain a generally similar performance as the ones reported in their paper. Our implementations of TD3BC, TD3BC-MCEP, AWAC, and AWAC-MCEP are based on (Kostrikov, 2022) framework. In all re-implemented/implemented methods, clipped double Q-learning (Fujimoto et al., 2018) is used. In TD3BC and TD3BC-MCEP, we keep the state normalization proposed in (Fujimoto & Gu, 2021) but other algorithms do not use it. For EQL and DQL, we use their official implementation and DQL-MCEP is also built upon the released codebase Wang et al., 2023.

The hyper-parameters used in the experiments are listed in Table 2.

| batch size | BC | IQL | AWAC | AWAC-MCEP | TD3BC | TD3BC-MCEP |
|---|---|---|---|---|---|---|
| actor LR | 1e-3 | 3e-4 | 3e-5 | 3e-5 | 3e-4 | 3e-4 |
| actor^e LR | | - | | 3e-5 | - | 3e-4 |
| critic LR | - | 3e-4 | | | | |
| $V$ LR | - | 3e-4 | - | | | |
| actor/critic network | (256, 256) | | | | | |
| discount factor | 0.99 | | | | | |
| soft update $\tau$ | - | 0.005 | | | | |
| dropout | 0.1 | - | | | | |
| Policy | | TanhNormal | | | Deterministic | |
| MuJoCo Locomotion | | | | | | |
| $\tau$ for IQL | - | 0.7 | - | | | |
| $\lambda / \tilde{\lambda}$ | - | $1/\lambda = 3$ | 1.0 | | - | |
| $\lambda^e$ | - | | 0.6 | | - | |
| $\alpha / \tilde{\alpha}$ | - | | | | 2.5 | |
| $\alpha^e$ | - | | | | 10.0 | |

Table 2: Hyper-parameters.

## A.2 Data collection and Hyperparameters tunning for humanoid tasks

**Hyperparameters.** In this experiment, we select Top-10 Behavior cloning, TD3BC and IQL as our baselines. For Top-10 Behavior cloning, only $10\%$ data of highest returns are selected for learning. For TD3BC, we searched the hyperparameter $\alpha = \{0.1, 0.5, 1.0, 2.0, 3.0, 4.0, 5.0\}$. For IQL, we searched the expectile hyperparameter $\tau = \{0.6, 0.7, 0.8, 0.9\}$ and the policy extraction hyperparameter $\lambda = \{0.1, 1.0, 2.0, 3.0\}$. For CRR, we tune the advantage coefficiency $\beta = \{0.1, 0.6, 0.8, 1.0, 1.2, 5.0\}$. For TD3BC-MCEP, we searched the $\tilde{\alpha} = \{0.1, 0.5, 1.0, 2.0, 3.0\}$ and $\alpha^E = \{3.0, 4.0, 5.0, 10.0\}$. The final selected hyperparameters are listed in Table 3. For CRR, we implement the CRR exp version based on (Hoffman et al., 2020). This version is considered as it outperforms other baselines in (Wang et al., 2020) in complex environments such as humanoid. We also applied *Critic Weighted Policy* as well as an argmax version of it (CRR-argmax). These design options result in CRR, CRR-CWP and CRR-Argmax variants. In Figure 6, we report the most performant CRR variant for each task. Among all its variants, CRR-Argmax shows better performance in both the *medium* and the *medium-replay* while CRR performs the best in the *medium-expert* task.

| batch size | BC | IQL | TD3BC | TD3BC-MCEP |
|---|---|---|---|---|
| actor LR | 1e-3 | 3e-4 | 3e-4 | 3e-4 |
| actor^e LR | | - | - | 3e-4 |
| critic LR | - | | 3e-4 | |
| $V$ LR | - | 3e-4 | | - |
| actor/critic network | | | (256, 256) | |
| discount factor | | | 0.99 | |
| soft update $\tau$ | - | | 0.005 | |
| dropout | 0.1 | | - | |
| Policy | TanhNormal | | Deterministic | |
| *Humanoid-medium-v2* | | | | |
| $\tau$ for IQL | - | 0.6 | | - |
| $\lambda/\tilde{\lambda}$ | - | 1 | | - |
| $\alpha/\tilde{\alpha}$ | | - | 1 | 0.5 |
| $\alpha^e$ | | - | | 3 |
| *Humanoid-medium-replay-v2* | | | | |
| $\tau$ for IQL | - | 0.6 | | - |
| $\lambda/\tilde{\lambda}$ | - | 0.1 | | - |
| $\alpha/\tilde{\alpha}$ | | - | 0.5 | 1.0 |
| $\alpha^e$ | | - | | 10 |
| *Humanoid-medium-expert-v2* | | | | |
| $\tau$ for IQL | - | 0.6 | | - |
| $\lambda/\tilde{\lambda}$ | - | 0.1 | | - |
| $\alpha/\tilde{\alpha}$ | | - | 2 | 0.5 |
| $\alpha^e$ | | - | | 3 |

Table 3: Hyper-parameters.

**Humanoide Data Collection.** In the table below, we provide details of the collected data.

| Task | # of trajectories | # of samples | Mean of Returns |
|---|---|---|---|
| Humanoid Medium | 2488 | 1M | 1972.8 |
| Humanoid Medium Replay | 3008 | 0.502M | 830.2 |
| Humanoid Medium Expert | 3357 | 1.99M | 2920.5 |

Table 4: Dataset statistics for humanoid offline data.

## A.3 Task specific parameters for TD3BC and TD3BC-MCEP

To investigate the optimal policy constraint strengths, we search this hyperparameter for TD3BC and TD3BC-MCEP. Their optimal values and the corresponding performance improvement are vi-

sualized in Figure 9. As we observed, in 7 of the 9 tasks, the optimal policies found by TD3-MCEP outperform optimal policies found by TD3BC. In all *medium* tasks, though the optimal constraint values are the same for TD3BC and TD3BC-MCEP, TD3BC-MCEP outperformance TD3BC. This is benefitted by that relaxing the constraint of evaluation policy does not influence the value estimate. For TD3BC, milder constraint might cause unstable value estimate during training. In all *medium-replay* tasks, we found optimal constraints for TD3BC-MCEP are milder than TD3BC, which verifies the requirements of milder constraints 5.2.

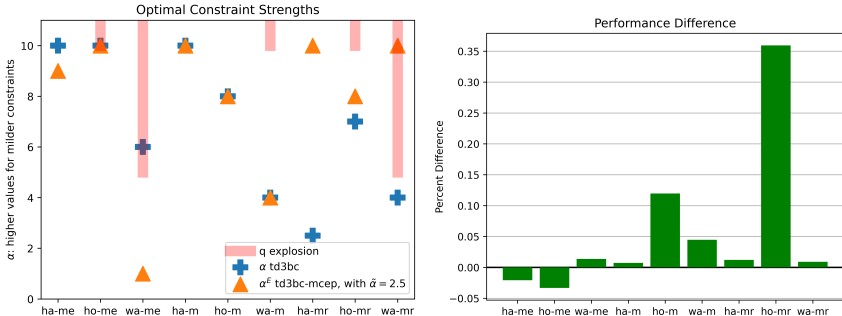

Figure 9: **Left:** Optimal $\alpha^E$ values for the evaluation policy of TD3BC-MCEP, with a fixed $\alpha = 2.5$ for the target policy. Optimal $\tilde{\alpha}$ values for TD3BC. Red areas indicate the $\alpha$ values for TD3BC that raises Q-value explosion (in one or more training of a 5-seed training). **Right:** Performance difference between the evaluation policy of TD3BC-MCEP and the actor of TD3BC, using the $\alpha^E$ ($\alpha$) values shown in the left figure.

### A.4 AN INVESTIGATION OF OTHER METHODS FOR INFERENCE-TIME POLICY IMPROVEMENT

The MCEP aims to improve the inference time performance without increasing the Bellman estimate error. Previous works also propose to use the on-the-fly inference-time policy improvement methods. For example, (Wang et al., 2020) proposes the *Critic Weighted Policy (CWP)*, where the critic is used to construct a categorical distribution for inference-time action selection. Another simple method is selecting the action of the largest Q values, namely **Argmax**. In this section, we compare the performance of TD3BC and TD3BC-MCEP under different test-time policy improvement methods.

The results are presented in Table 5 and 6. Both the Argmax and the CWP methods select an action from an action set. We generate this action set by adding Gaussian noise to the outputs of the deterministic policy. The *std* is the noise scale and N is the size of this action set. From the results, we observe that CWP and Argmax help improve the performance of both the TD3BC and TD3BC-MCEP. It is worth noting that, in *medium* task, the Argmax method improves the TD3BC to the same level as TD3BC-MCEP. But in *meidum-replay* and *medium-expert* tasks, the improved performances are still worse than the TD3BC-MCEP. On TD3BC-MCEP, applying Argmax and CWP further improves policy performances.

In conclusion, the inference-time performance could be improved by utilizing the methods mentioned above but MCEP shows a more significant policy improvement and does not show conflict with these on-the-fly methods.

Table 5: TD3BC with inference-time policy improvement. The original policy has returns 2483.9, 965.4 and 3898.2 for medium, medium-replay and medium-expert, respectively.

| Game | N\std | Argmax | | | | CWP | | | |
|---|---|---|---|---|---|---|---|---|---|
| | | 0.01 | 0.02 | 0.05 | 0.1 | 0.01 | 0.02 | 0.05 | 0.1 |
| medium | 20 | 2462.4 | 2944.5 | 3098.9 | **3511.1** | 2441.4 | 2689.8 | 2755.4 | 3113.2 |
| | 50 | 2564.2 | 2836.0 | 2956.6 | 3156.3 | 2159.8 | 2588.6 | 2462.1 | **2839.6** |
| | 100 | 2857.0 | 2369.8 | 3122.0 | 3266.8 | 2607.6 | 2665.8 | 2722.5 | 2584.1 |
| medium-replay | 20 | 895.3 | 1042.3 | 1136.7 | 1524.1 | 973.9 | 931.9 | 932.2 | **1242.7** |
| | 50 | 994.6 | 976.7 | 1160.2 | **1664.9** | 974.7 | 1030.3 | 1002.1 | 1171.1 |
| | 100 | 971.7 | 1049.0 | 1232.2 | 1574.7 | 874.2 | 1023.5 | 973.0 | 1232.9 |
| medium-expert | 20 | 3861.7 | 4068.4 | 4131.0 | 4585.3 | 4181.1 | 4478.3 | 3904.0 | 3636.7 |
| | 50 | 4460.6 | 4012.2 | **4612.9** | 4603.0 | 3987.0 | 4068.9 | 3995.1 | **4214.7** |
| | 100 | 4130.8 | 4141.7 | 4158.3 | 4421.4 | 4145.3 | 3634.6 | 3933.9 | 3788.3 |

Table 6: TD3BC-MCEP with inference-time policy improvement. The original policy has returns 2962.8, 4115.6 and 4829.2 for medium, medium-replay and medium-expert, respectively.

| Game | N\std | Argmax | | | | CWP | | | |
|---|---|---|---|---|---|---|---|---|---|
| | | 0.01 | 0.02 | 0.05 | 0.1 | 0.01 | 0.02 | 0.05 | 0.1 |
| medium | 20 | 2368.2 | 2871.4 | 2924.1 | 3392.8 | 2670.5 | 2710.0 | 3146.5 | 2987.9 |
| | 50 | 2822.1 | 3046.3 | 3283.9 | **3861.7** | 2612.9 | 2787.1 | 2718.9 | 2841.7 |
| | 100 | 3405.1 | 2808.2 | 3264.5 | 3751.3 | **3003.3** | 2896.8 | 2748.9 | 2727.2 |
| medium-replay | 20 | **4277.3** | 4071.5 | 4092.7 | 4253.4 | 4033.1 | 4200.0 | 4254.4 | 4167.0 |
| | 50 | 4225.5 | 4159.7 | 4028.4 | 4210.8 | 4135.5 | 4219.1 | **4375.7** | 4230.3 |
| | 100 | 4190.3 | 3966.4 | 4138.4 | 4270.1 | 4328.5 | 4266.9 | 4275.7 | 4142.5 |
| medium-expert | 20 | 4752.8 | 4956.7 | 4880.3 | 4887.1 | 4736.4 | 4710.8 | 4748.7 | 4942.1 |
| | 50 | 4930.7 | **5018.2** | 4614.2 | 4899.9 | **5053.4** | 5001.4 | 4808.8 | 4670.6 |
| | 100 | 4616.8 | 4800.9 | 4700.9 | 4648.0 | 4588.1 | 4770.6 | 4934.3 | 4855.3 |

## A.5 DESIGN OPTION OF EVALUATION POLICY UPDATE

As the evaluation policy is not involved in the actor-critic's iterative update, one might want to update the evaluation afterward, i.e., update the evaluation from the critic after the actor-critic converges, namely **afterward updates**. While this is a valid design option, our method simultaneously updates the target policy and the evaluation (**simultaneous updates**). In this manner, their updates can be parallelized and no further time is required based on the actor-critic training. Figure 10 and 11 present the convergence for these two design options. From the results, we observe a faster convergence of afterward updates in some tasks. However, there are also many tasks where the afterward updates method converges after a million steps. For methods of slow training (e.g. DQL), this afterward training time becomes significant.

## A.6 FULL RESULTS FOR ESTIMATED Q VALUES OF THE LEARNED EVALUATION POLICIES

Figure 12 and Figure 13 show the visualization of the estimated Q values achieved by the target policy and evaluation policy.

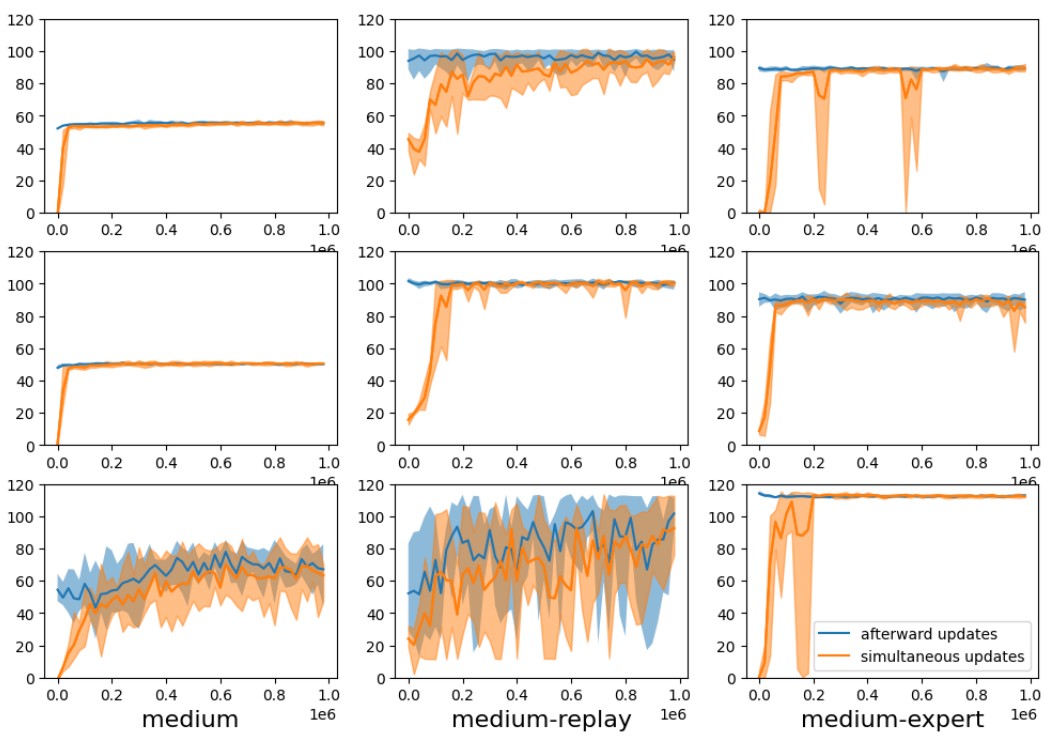

Figure 10: The comparison of *simultaneous updates* and *afterward updates* for the evaluation for TD3BC-MCEP. **First row:** *halfcheetah*. **Second row** *hopper*. **Third row:** *walker2d*.

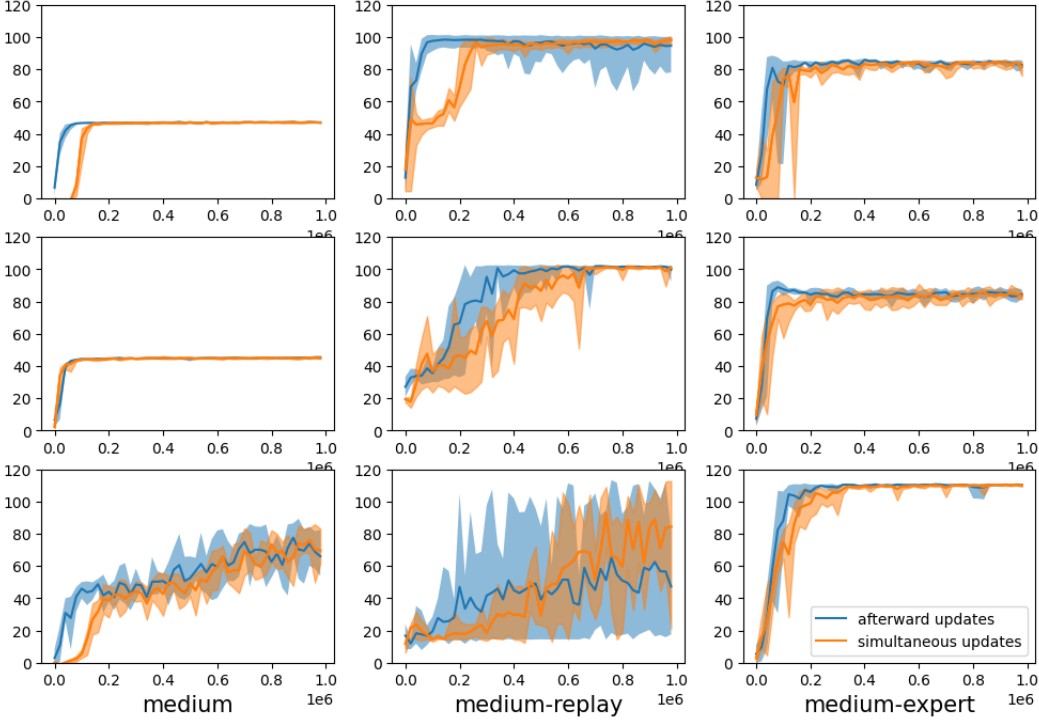

Figure 11: The comparison of *simultaneous updates* and *afterward updates* for the evaluation for AWAC-MCEP. **First row:** *halfcheetah*. **Second row** *hopper*. **Third row:** *walker2d*.

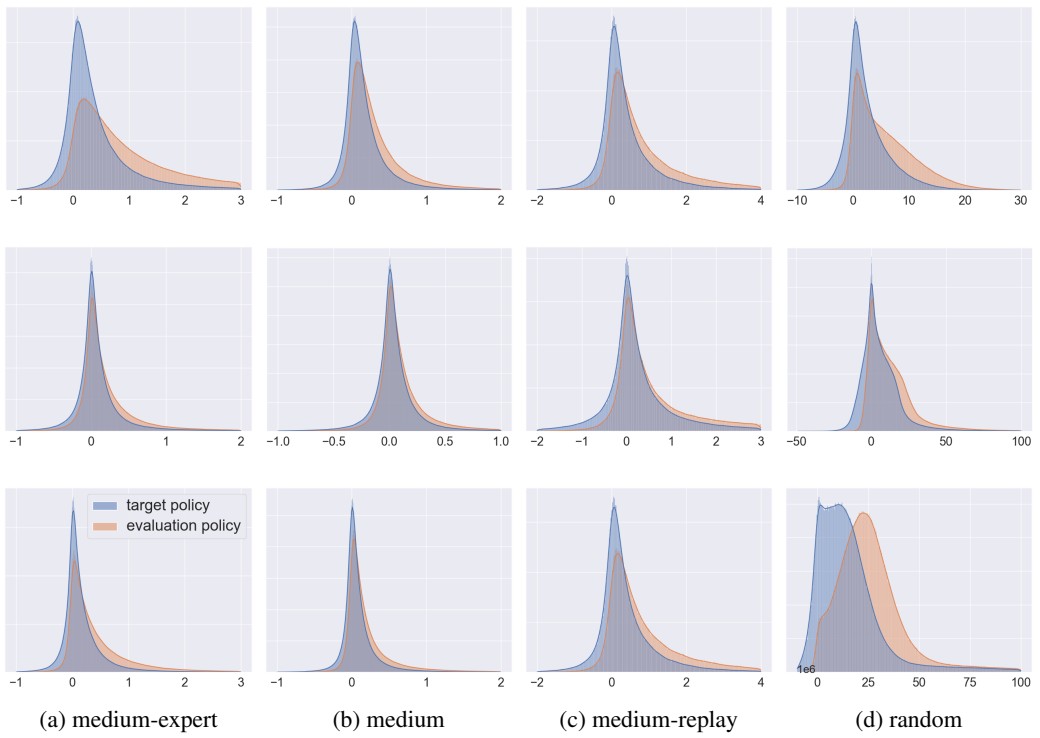

Figure 12: TD3BC-MCEP. **First row:** *halfcheetah*. **Second row** *hopper*. **Third row:** *walker2d*.

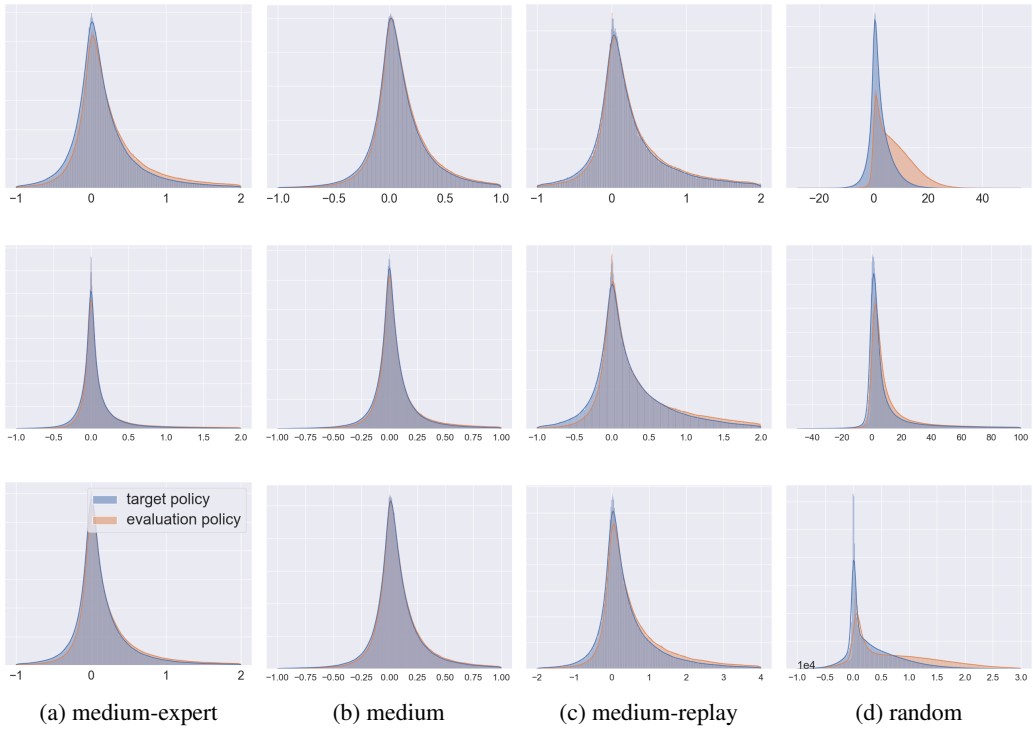

Figure 13: AWAC-MCEP. **First row:** *halfcheetah*. **Second row:** *hopper*. **Third row:** *walker2d*.

