# OpenReview forum: "Mildly Constrained Evaluation Policy for Offline Reinforcement Learning"
_ICLR.cc/2024/Conference — ICLR 2024 Conference Withdrawn Submission_

### Official Review · Reviewer_WVL4 · 2023-10-20

**Soundness:** 2 fair
**Presentation:** 3 good
**Contribution:** 3 good
**Rating:** 5
**Confidence:** 4

**Summary:**

This paper focuses on the trade-off between value estimation stability and performance improvement caused by in offline RL. The authors propose a new algorithm by introducing a new mildly constrained policy to obtain both stable value estimation and good evaluation performance. The authors test their method on D4RL mujoco tasks to verify its effectiveness.

**Strengths:**

- The paper is clearly written and easy to follow.
- The trade-off between stability of value learning and policy improvement is important and not well-studied by previous work.
- The experiment part adequately explains how the policy constraint influences the evaluation and policy evaluations, which validates the motivation of this paper.

**Weaknesses:**

- The authors claim that a mild-constrained evaluation policy improves the final performances, but its effectiveness is questionable.
    - The improvement may be attributed to the policy constraint strengths of original method are not well selected. E.g., in fig.4, the performance of TD3BC on hopper-m can achieve >80 with $\alpha=10$. If we use this value as the baseline, then the improvement of the proposed method is actually limited. This also happens on other two settings plotted in fig.4. Meanwhile, if the original policy constraint strengths are suitable, a milder constraint in MCEP may actually degrade the performances (e.g., TD3+BC on medium-expert tasks).
    - For DQL and DQL-MCEP, there are no remarkable differences on most tasks. So why MCEP is effective on some baselines but helps little on others?
    - Based on the above analysis, we can find whether the additional evaluation policy improves the performances heavily depends on the strength of policy constraint in original baseline. MCEP can achieve better with better hyper-parameter, which is actually infeasible in offline RL setting, however.
    - Although the authors give an ablation study in sec. 5.4, the improvement of MCEP is not very significant and the results seem to be inconsistent with previous figures and tables (See questions).
- The authors only test their methods on mujoco tasks. To comprehensive verify the advantages of proposed method, more experiment results (e.g., on maze/kitchen/adroit) are needed.

Minor issues:
- There are two `3)` in the first paragraph of sec.5.
- In the first paragraph in page 9, $\\tilde{\\alpha}, \\tilde{\\lambda}$ instead of $\\tilde{alpha}, \\tilde{lambda}$.

**Questions:**

- Which level are you using in fig.7? If it is "-medium", why are the performances of TD3BC with $\alpha=2.5$ and TD3BC-MCEP very different from the values reported in table 1? The hyper-parameters should be the same.

---

> ### Author Response · Authors · 2023-11-14
> **A Response to Reviewer WVL4**
>
> We thank the reviewer WVL4 for their review of our manuscript.
>
> We appreciate reviewer WVL4 for investigating the experiment details but we find one overlooked setting of our experiment could guide to some misunderstanding.
>
> >The improvement may be attributed to the policy constraint strengths of the original method are not well selected…
>
> We use the setting of “similar hyparameters for all locomotion domains” as mentioned in the paper (line 3, Section 5.3 *“For D4RL datasets, similar hyperparameters are used”*). This setting is widely used in current offline RL literature (Fujimoto et al., 2020, Kostrikov et al., 2021, Hansen-Estruch et al., 2023, Wang et al., 2023, etc). Table 1 reports the best performance under this setting, both for the baseline and for our methods. With this evaluation protocol, **the hyperparameters for original methods are well-selected.**
>
> In, fig.4, we investigate multiple strengths for the original methods. It is possible that one constraint strength suits a specific task best but does not derive the best general performance. We have an experiment that compares task-wise best hyper-parameters, which can be found in Section A.3 of the Appendix, where in 7 out of 9 tasks, the MCEP outperforms the original algorithm.
>
> >For DQL and DQL-MCEP, there are no remarkable differences on most tasks…
>
> One of the motivations for using MCEP is to reduce the Bellman estimate error brought by OOD actions. Conventional methods use a Gaussian policy that raises many OOD actions under multi-modal distribution (Cai et al., 2022, Wang et al., 2023, Hansen-Estruch et al., 2023). MCEP significantly eliminates these OOD actions by using a target policy of strong constraint, which leads to significant performance improvement.
>
> DQL utilizes diffusion policies that effectively learn the multi-modal distribution. Hence the Bellman estimate error is reduced. In this case, MCEP can still improve its general performance by optimizing the evaluation policy towards higher-value OOD actions further.
>
> >Based on the above analysis, we can find whether...
>
> The performance of MCEP depends on the stable value estimate, which is related to the hyperparameter of the base algorithm. In our experiments, we use paper-recommended hyperparameters for the target policy.
>
> In Section A.3 (fig. 9), under the recommended hyperparameter (same value for all tasks) for the target policy, only tuning the hyperparameter for the MCEP enables it to outperform the base algorithm that uses task-specific best hyperparameter in 7/9 tasks. This means that it is not necessary to have the best hyperparameter of the base algorithm for the MCEP to achieve better performance.
>
> >Although the authors give an ablation study in sec. 5.4, the improvement of MCEP is not very significant and the results seem to be inconsistent with previous figures and tables (See questions).
>
> Under the setting of "similar hyperparameters for all locomotion tasks", all experiment results stay consistent. The improvement of MCEP is significant as under a widely used evaluation protocol, it empowers conventional algorithms to achieve near SOTA performance and provide a further improvement for the SOTA method.
>
> >The authors only test their methods on mujoco tasks…
>
> As this is asked by reviewer **r8JF, VtJK, WVL4 (3/4 of the reviewers),** we answer this question in the general review that can be found at the top of this page.
>
> >Which level are you using in fig. 7? …
>
> We apologize for the confusion. Fig. 7 presents the average performance of all versions of datasets for each domain. E.g. the bars above the “halfcheetah” presents the average scores of the “medium”, “medium-replay” and “medium-expert” datasets. With this setting, it is reasonable for Fig.7 and Table 1 to have different scores. We add this explanation in Section 5.4 “The scores for different datasets are grouped for each domain.”
>
> Please let us know whether our answers address all your concerns. We are more than happy to answer more if you have further questions and take suggestions to further improve our work.
>
> ## References
>
> Fujimoto, S. and Gu, S.S., 2021. A minimalist approach to offline reinforcement learning. *Advances in neural information processing systems*, *34*, pp.20132-20145.
>
> Kostrikov, I., Nair, A. and Levine, S., 2021. Offline reinforcement learning with implicit q-learning. *arXiv preprint arXiv:2110.06169*.
>
> Hansen-Estruch, P., Kostrikov, I., Janner, M., Kuba, J.G. and Levine, S., 2023. Idql: Implicit q-learning as an actor-critic method with diffusion policies. *arXiv preprint arXiv:2304.10573*.
>
> Wang, Z., Hunt, J.J. and Zhou, M., 2022. Diffusion policies as an expressive policy class for offline reinforcement learning. *arXiv preprint arXiv:2208.06193*.
>
> Cai, Y., et al, 2022, November. TD3 with Reverse KL Regularizer for Offline Reinforcement Learning from Mixed Datasets. In *2022 IEEE International Conference on Data Mining (ICDM)* (pp. 21-30). IEEE.

---

> > ### Comment · Reviewer_WVL4 · 2023-11-19
> >
> > Thanks for your clarifications on the settings and results in experiment part and now I am convinced with the results on mujoco tasks in paper. However, based on your reply and your discussion with other reviews, I still have following concerns:
> >
> > - The performances of MCEP on Antmaze/Adroit/kitchen are not good and cannot outperform the baselines. Although the authors gave some reasons, it shows that the improvement by MCEP is not applicable to other tasks, which limits the contribution of this paper.
> >
> > - Even on the mujoco tasks, the idea of MCEP may not be applicable/helpful to other existing offline methods: 1) the improvement on DQL is marginal; 2) the authors claim that MCEP cannot be used on CQL, which is strange because CQL also encounters the tradeoff between stable Q estimate and evaluation performance.

---

> > > ### Author Response · Authors · 2023-11-19
> > >
> > > Thank you for your reply.
> > >
> > > >The performances of MCEP on Antmaze/Adroit/kitchen...
> > >
> > > These datasets show a variaty challenges for offline RL, e.g. sparse rewards, limited data. Instead of pursuing superior algorithmic performance, we focus on an important fundamental problems for offline RL:
> > >
> > > **P1.** value learning
> > >
> > > **P2.** inference-time performance
> > >
> > > and provide insights by our empirical analysis. These insights include:
> > >
> > > - A further understanding to the drawback of policy constraints methods. Most of them treat P1. and P2. as a single problem. While these methods focus on addressing P1., they expect the solutions to solve P2. Our experiments 5.1 and 5.2 visualize this drawback.
> > > - A novel perspective to understand the modules in policy constraints methods (e.g. AWAC and TD3BC), showing that while the policy is not performed, its critic is well-learned.
> > > - We propose to separate P1. and P2. Under the policy constraints framework, we devise a simple yet general MCEP method to improve the inference-time performance. Although the improvement on DQL is not by a large margin, a more complex method based on our insight might obtain better performance.
> > >
> > > > 2) the authors claim that MCEP cannot be used on CQL, which is strange because CQL also encounters the tradeoff between stable Q estimate and evaluation performance.
> > >
> > > Pessimistic value methods such as CQL also have the same drawback by solving 1) and 2) with the same degree of conservative. How to address this drawback for pessimism methods is still an open problem (Hong et al 2022, Hong et al 2023). While the idea of separating P1. and P2. is general, please note that the MCEP is devised under the framework of policy constraints method, which is different from the pessimism methods.
> > >
> > > We hope our reply helps you to understand the contribution of this work. We are pleased to answer more questions.
> > >
> > > **References**
> > >
> > > Hong, J., Kumar, A. and Levine, S., 2022. Confidence-conditioned value functions for offline reinforcement learning. arXiv preprint arXiv:2212.04607.
> > >
> > > Hong, Z.W., Kumar, A., Karnik, S., Bhandwaldar, A., Srivastava, A., Pajarinen, J., Laroche, R., Gupta, A. and Agrawal, P., 2023. Beyond Uniform Sampling: Offline Reinforcement Learning with Imbalanced Datasets. arXiv preprint arXiv:2310.04413.

---

> > > > ### Author Response · Authors · 2023-11-22
> > > >
> > > > > it shows that the improvement by MCEP is not applicable to other tasks, which limits the contribution of this paper.
> > > >
> > > > To figure out whether the MCEP can work in complex domains and better address this concern, we conduct a new experiment on a diverse set of 16 robotic manipulation tasks (Jorge et al 2022, Hussing et al 2023): 16 tasks $\times$ 2-version datasets for each task.
> > > >
> > > > - Average Success (%) Rate among 16 robotic manipulation tasks
> > > > | Dataset      | BC       | CRR    | TD3BC           | IQL | TD3BC-MCEP |
> > > > | ------------ | --------- | ---------- | --------- | --------- | --------- |
> > > > | Medium |29.0 | 56.9| 45.9| 55.5  | **59.6**|
> > > > | Medium Replay |1.2 |11.9|15.0| 25.0 |  **26.4** |
> > > >
> > > > The above results present that the MCEP (based on TD3BC) improves the performance significantly in these complex tasks. For more details about their hyperparameters and the success rate for each task, please refer to the general response on the top of this page.
> > > >
> > > > We hope the new experiment results have solved your remaining concerns. Plese let us know if anything unclear.
> > > >
> > > > **References**
> > > >
> > > > Jorge A. et al., CompoSuite: A compositional
> > > > reinforcement learning benchmark. In 1st Conference on Lifelong Learning Agents, 2022a.
> > > >
> > > > Hussing, et al., 2023. Robotic Manipulation Datasets for Offline Compositional Reinforcement Learning. arXiv preprint arXiv:2307.07091.

---

### Official Review · Reviewer_px9k · 2023-10-20

**Soundness:** 4 excellent
**Presentation:** 3 good
**Contribution:** 2 fair
**Rating:** 5
**Confidence:** 4

**Summary:**

This paper studies the policy constraint methods in offline reinforcement learning. It takes an interesting idea: decoupling the constraint strength for stable value estimation and for policy learning. Specifically, they find that while we need a restrictive policy constraint to mitigate extrapolation error in value estimation, a milder constraint is allowed for policy learning. Thus, apart from the target policy used in actor-critic learning of standard offline RL, a mildly constrained evaluation policy (MCEP) is proposed to be separately learned with a more relaxed policy constraint. This paper instantiates MCEP with existing offline RL method TD3+BC, AWAC, and DQL and demonstrates improved performance.

**Strengths:**

1. An ingenious idea of decoupling constraint strengths for stable value estimation and for policy learning. Two kinds of distributional shifts, namely OOD actions during Bellman bootstrapping and deployment, to my knowledge, are rarely distinguished in offline RL literature.
2. The design of MCEP is simple and general for policy constraint methods.
3. Challenging humanoid tasks are introduced in the experiments, and MCEP demonstrates good performance.
4. The visualization of the toy example in Figure 2 illustrates the motivation well.

**Weaknesses:**

1. The most important finding of this paper, i.e. the difference between policy constraint strengths for stable value learning and for a performant evaluation policy, is validated empirically but lacks a theoretical analysis.
2. The improvement of MCEP upon DQL is limited, which doubts the benefits of MCEP for modern offline RL methods with better designs, such as ReBRAC.
3. The proposed MCEP is only applicable to policy constraint methods and does not outperform other kinds of sota offline RL methods, such as MCQ and EDAC.

**Questions:**

1. What do you mean by 'While the target policy may recover its performance by iterative policy improvement and policy evaluation, we observe that the evaluation policy may fail to do so.'
2. How does MCEP 'overcome this drawback' (state-agnostic constraint) discussed in Singh et al.?
3. As the authors claim in the introduction that the toy maze experiments 'validate the finding of (Czarnecki et al., 2019)', I recommend also mentioning Czarnecki et al. in Section 5.1.

Typos:

- Section 4.3 DQL WITH MCEP should be a paragraph in parallel with TD3BC with MCEP and AWAC with MCEP.
- $C\left(\pi_{\beta, \pi^E}\right)$ should be $C\left(\pi_{\beta}, \pi^E\right)$ in the Equation (10)
- There seem to be some explanation sentences missing after 'We next introduce the policy improvement step for the evaluation policy' and Equation (10).
- The caption of Figure 6 is incorrectly the same as that of Figure 7.

---

> ### Author Response · Authors · 2023-11-14
> **A Response to Reviewer px9k**
>
> We appreciate reviewer px9k for their review and kind comments on our manuscript.
>
> >1 The most important finding of this paper, is validated empirically but lacks a theoretical analysis …
>
> As you mentioned in your comment, *the two kinds of distributional shifts are rarely distinguished in offline RL papers*. This differentiation is fundamental but important for offline RL study. Our work focuses on empirical analysis to reveal this differentiation and proposes a general method to address them separately. We are looking forward to building up the theory by separating these distributional shifts in the near future.
>
> >2.The improvement of MCEP upon DQL is limited, which doubts the benefits of MCEP for modern offline RL methods with better designs, such as ReBRAC.
>
> One of the motivations for using MCEP is to reduce the Bellman estimate error brought by OOD actions. Conventional methods use a Gaussian policy that raises many OOD actions under multi-modal distribution. MCEP significantly eliminates these OOD actions and leads to significant performance improvement. DQL utilizes diffusion policies that effectively learn the multi-modal distribution. Hence the Bellman estimate error is reduced. In this case, MCEP can still improve its general performance by optimizing the evaluation policy towards higher-value OOD actions further.
> **we respectfully argue that our contributions do not emphasize strong algorithmic performance, but are two-fold, as concluded below:**
>
> 1) We propose to distinguish the distributional shifts for Bellman estimate error reduction and the inference-time performance. Existing policy constraints methods (Kumar et al 2019, Nair et al 2020, Wang et al., 2023 etc) focus on addressing the first type but wish to achieve the second type.
>
> 2) We propose to address these two types of distributional shifts separately and propose a general algorithm that learns two policies with different policy constraints. Our experiments on both D4RL locomotion and the challenging humanoid illustrate the performance improvement brought by the MCEP.
>
> >3. The proposed MCEP is only applicable to policy constraint methods and does not outperform other kinds of sota offline RL methods, such as MCQ and EDAC.
>
> One of the contributions of this work is to distinguish two types of distributional shifts. Our work investigates this phenomenon in policy constraint strengths, hence the resulting method is applied to policy constraints methods, which is a large group of offline RL methods. We are looking forward to investigating this phenomenon in other methods in the future.
>
> Our comparison includes EQL and DQL which are also highly-performed SOTA algorithms. As both the MCQ and EDAC reported performances with task-specific hyper-parameters while we use “similar hyperparameters to locomotion tasks”, the reported performance under different evaluation protocols does not provide a straightforward comparison.
>
> >1. What do you mean by 'While the target policy may recover its performance…
>
> Unstable Q learning means that the value of a bad-performed action could sometimes be arbitrarily high but can later back to an accurate estimate when 1) this action is queried during TD learning and 2) the dataset distribution covers this state-action. The target policy might learn this bad-performed action when it’s of arbitrarily high value. But the critic will query this target policy during the TD learning, hence the inaccurate value estimate for this action is fixed. For the evaluation policy, it can learn a different bad-performed action from the target policy’s. But the critic never queries the evaluation policy, hence the value estimate for this action is always arbitrarily high and the behavior of the evaluation policy does not recover to good behavior.
>
> >2. How does MCEP 'overcome this drawback' (state-agnostic constraint) discussed in Singh et al.?
>
> Signh et al studies the datasets where the behavior diversity varies among different states. The appropriate constraint strength for a state of low diversity may be too restrictive for the policy to learn behavoir of another state of high diversity. A milder constraint strength enables the policy to learn diverse behavior but increase the Bellman estimate error on the state of low diversity. The MCEP uses the target policy to obtain stable Bellman estimate and the evaluation policy to learn more diverse behavior.
>
> >3. I recommend also mentioning Czarnecki et al. in Section 5.1
>
> We added these lines in Section 5.1 *“This approach of utilizing the value function of the imperfect teacher policy is originally suggested by Czarnecki et al. (2019)”*. Thank you for your thoughtful suggestion.
>
> >Typos
>
> We fixed all the typos in the updated manuscript.
>
> Please let us know whether our answers address all your concerns. We are more than happy to answer more if you have further questions and take suggestions to further improve our work.

---

> > ### Comment · Reviewer_px9k · 2023-11-15
> >
> > Thanks for the detailed response!
> >
> > **On the limited improvement upon DQL**
> >
> > I am sorry that the response does not fully address my previous concern. I noticed that Reviewer WVL4 also mentioned the marginal performance difference between DQL and DQL-MCEP. The average score of DQL is 85+-1.6, and that of DQL-MCEP is 86.2+-1.2. Thus, the difference is not significant.  As the authors respond, it is because DQL utilizes diffusion policies that effectively learn the multi-modal distribution, which can effectively reduce Bellman estimate error. Given a powerful distribution estimator is sufficient to gain a performant policy, how much improvement room is there for a more mildly constrained evaluation policy, which still needs to be limited to avoid extreme OOD action? This is why I regret that there is no rigorous theoretical analysis.
> >
> > Looking forward to more insights from the authors.
> >
> > **(Minor) On the state-agnostic constraint**
> >
> > I understand that a complex behavior distribution demands different strengths for different states. But MCEP still adopts a state-agnostic constraint, right? It only utilizes two kinds of constraints, but each constraint the policy with an equal strength across states.

---

> > > ### Author Response · Authors · 2023-11-16
> > >
> > > >  how much improvement room is there for a more mildly constrained evaluation policy, which still needs to be limited to avoid extreme OOD action?
> > >
> > > If we understand correctly, a powerful distribution estimator under this context means that this policy is sufficient to approximate the dataset distribution. Hence during the actor-critic, it does not raise too many OOD actions and the Q becomes stable. We next explain why this estimator could still be problematic during offline learning.
> > >
> > > To explain the insight, we define the “edge” of the dataset distribution as “actions beyond this edge will cause severe Bellman approximate error”. In cases when some actions are exactly at the edge of the dataset distribution but are of high value, the policy (actor) distribution would move towards these edge actions. In one case, the mean of the policy distribution becomes this high-value action (to maximize its probability). However, as long as the standard deviation of the policy distribution is not zero, and because this action is at the edge of the dataset distribution, beyond-edge actions would be sampled, causing severe Bellman estimate error. With an MCEP, the **target policy** does not need to put its mean to this action but only needs to cover this action (to support the critic to learn the value of this edge action). The **evaluation policy**, instead, is free to move towards this action (maybe its mean becomes this edge action).
> > >
> > > >But MCEP still adopts a state-agnostic constraint, right?
> > >
> > > Thanks for asking. Your understanding is correct.
> > >
> > > >It only utilizes two kinds of constraints, but each constraint the policy with an equal strength across states.
> > >
> > > In Singh et al, section 3.1 mentions *“the learned policies match the random behavior of the dataset actions too closely in the wider rooms, and therefore are unable to make progress towards the Goal position. This is a direct consequence of enforcing too strong of a constraint on the learned policy to stay closely in the wider rooms”*
> > >
> > > With an MCEP, the *learned policy (actor)* that stays too close to the dataset becomes a problem no more, as we do not need it to perform well. We rephrase the corresponding text in the manuscript *“overcomes the drawback”* to *“overcomes the overly restrictive constraint problem (Singh et al., 2022) by using an extra evaluation policy.”*

---

> ### Comment · Reviewer_px9k · 2023-11-17
>
> Thanks for the prompt reply.
>
> Let me elaborate on my concern in the 'improvement room.' According to my understanding and the author's response, there is an "edge” in distribution avoiding severe Bellman approximate errors, and also an edge avoiding OOD states due to OOD actions during evaluation, which is wider than the former one and allows more performant learned policies. Given that the former edge can be pushed approaching the latter one with powerful policy distribution classes (e.g., beyond isotropic deviations) or better policy constraint formulations, my question is, how much room is still in between them? This determines the improvements that MCEP can give to base offline RL methods. Regretfully, the empirical results of MCEP upon DQL seem to suggest the improvement room is small for modern offline RL methods.
>
> Please correct me if there is any misunderstanding.

---

> > ### Author Response · Authors · 2023-11-17
> >
> > We are grateful for your precise comprehension. We also find it a potential direction to further investigate this improvement room, which includes:
> >
> > a) Theoretically, building up theoretical results to quantify this improvement room
> >
> > b) Empirically, devising more complex methods based on the insight of separating:
> >
> > - **P1.** value learning and
> >
> > - **P2.** inference-time performance.
> >
> > This work focuses on empirical analysis and these results provide insights including:
> >
> > - A further understanding of the drawback of policy constraints methods. Most of them treat **P1.** and **P2.** as a single problem. While these methods focus on addressing **P1.,** they expect the solutions to solve **P2.** Our experiments 5.1 and 5.2 visualize their drawback.
> > - A novel perspective to understand the modules in policy constraints methods (AWAC and TD3BC), showing that while the policy is not performed, its critic is well-learned.
> > - We propose to separate **P1.** and **P2.** Under the policy constraints framework, we devise a simple yet general MCEP method to improve the inference-time performance. Although the improvement on DQL is not by a large margin, a more complex method based on our insight might obtain better performance.
> >
> > We hope our clarification helps further understanding of the contribution of this work. Are there any more questions we can help answer?

---

### Official Review · Reviewer_VtJK · 2023-10-27

**Soundness:** 2 fair
**Presentation:** 3 good
**Contribution:** 2 fair
**Rating:** 6
**Confidence:** 3

**Summary:**

In policy constraint offline reinforcement learning (RL) algorithms, it is a common practice that the constraints for both value learning and test time inference are the same. This paper argues that such a paradigm may hinder the performance of the agent during test time inference. To address this issue, they propose the Mildly Constrained Evaluation Policy (MCEP) for test time inference. The idea is quite simple and the implementation is also easy. MCEP has the same objective function as the policy trained during the offline phase, but it does not participate in the policy evaluation phase. The authors show that by doing so, the performance of the offline RL agents can be improved.

**Strengths:**

# Strengths

This paper is generally well-written, and the logical flow is clear. I would say this paper is also well-motivated and proposes an interesting test-time inference algorithm. The resulting method is very simple, and the authors provide some figures and a toy example to illustrate to the readers the key idea behind their method, which I personally like very much. The authors combine their method with three off-the-shelf offline RL algorithms, and conduct some experiments on the D4RL locomotion datasets. The authors also conduct experiments on the Humanoid datasets, where the authors collect the corresponding static datasets by themselves. One can observe performance improvement by building MCEP upon numerous base algorithms. To summarize, the strengths and the advantages of this manuscript are

- this paper is well-written with a clear logic flow

- the core idea and the resulting method of this paper is quite simple and easy to implement

- the improvements from the proposed method are significant on many base algorithms

- the authors provide source codes, and I believe that the results presented in this paper are reproducible

**Weaknesses:**

# Weaknesses

I think the submission has the following potential flaws

- (major) limited evaluation. Though the authors combine their proposed MCEP method with three offline RL algorithms, they only evaluate them on locomotion tasks, which are actually simple and easy to get a high return. So, my question is, can the proposed method benefit other domains like antmaze, kitchen, and adroit? These domains are known to be more challenging than the MuJoCo tasks. I strongly believe that the empirical evaluations on these domains are critical to show the effectiveness and advantages of the proposed methods. If the proposed methods fail in these domains, I also expect possible explanations from the authors. This paper feels quite incomplete without the experiments on these domains.

- (major) It turns out that the hyperparameter selection counts in MCEP. Based on the empirical results in Section 5.4, TD3BC-MCEP and AWAC-MCEP are slightly sensitive to the hyperparameters. This may cause issues when using the MCEP in practice. Can the authors further explain this phenomenon and are there any ways that we can get rid of it?

- (minor) inconsistent abbreviation for some of the algorithms, e.g., the authors write TD3-BC in the first few paragraphs while using TD3BC later. This is not a big issue and can be easily fixed, please check your submission for potential similar issues.

I will be happy to update my score if the concerns are addressed during the rebuttal or in the revised manuscript.

**Questions:**

It seems your method is not restricted to the policy constraints offline RL methods, can your method be applied to value-based offline RL algorithms like CQL? I would expect explanations from the authors if CQL-MCEP fails and underperforms vanilla CQL.

---

> ### Author Response · Authors · 2023-11-14
> **A Response to Reviewer VtJK**
>
> We thank the reviewer **VtJK** for their thorough feedback on our work.
>
> > (major) Limited Evaluation …
>
> About the evaluation on antmaze, kitchen and adriot, as this is asked by reviewers **r8JF, VtJK, WVL4 (3/4 of the reviewers),** we answer this question in the general review that can be found at the top of this page.
>
> > (major) It turns out that the hyperparameter selection…
>
> **An explanation for this phenomenon:** Consistent with the results of section 5.4 (Figure 8), a strict constraint hinders the policy from improving beyond the datasets so may show poor performance. With milder constraints, the policy is able to utilize the estimated q function and take some OOD actions of high value during the inference so it achieves better performance over the dataset.
>
> **Suggestions about the hyperparameter selection:** Based on the results of Section 5.4, we suggest using the recommended constraint strength from the base algorithm paper for the target policy. This ensures a stable q estimate. Then tune the evaluation policy constraint strength to wilder values. Our experiment also follows this tuning protocol for all MCEP methods.
>
> > (minor) inconsistent abbreviation
>
> we unify them by using TD3BC (including the figures) in the newly submitted manuscript. Thanks for your thoughtful suggestion.
> >can your methods be applied to … like CQL?
>
> One of the motivations for using MCEP is to avoid the Bellman estimate error brought by the actor with milder constraints. In the pessimism methods such as CQL, the Q estimate error is reduced by pushing the Q values down for OOD actions. The policy in CQL does not need the policy constraint therefore milder constraint (or non-constraint) policy does not exacerbate the Bellman estimate error and thus the MCEP does not help in these methods.
>
> Please let us know whether our answers address all your concerns. We are more than happy to answer more if you have further questions and to take suggestions to improve our work.

---

> > ### Comment · Reviewer_VtJK · 2023-11-16
> >
> > My concerns seem not to be addressed.
> >
> > The authors write that MCEP fails to outperform baselines on Adroit, AntMaze, and Kitchen. Meanwhile, MCEP cannot help value-pessimism methods like CQL. Can you provide detailed numerical results? It should not be difficult to run MCEP on these datasets.
> >
> > So, MCEP can only aid policy constraint methods? Can it benefit IQL as well?

---

> ### Author Response · Authors · 2023-11-16
>
> Thanks for your thoughtful discussion.
>
> > Can you provide detailed numerical results?
>
> We provide the results of two tasks from Adroit below. The results were taken from the last step of the training, with the average return and standard error from 5 random seeds.
>
> | Dataset      | BC       | CQL    | IQL           | TD3BC | TD3BC-MCEP | AWAC | AWAC-MCEP |
> | ------------ | -------- | ---------- | --------- | --- | ---- | --- | -------- |
> | pen-human |$76.8\pm4.8$ | $37.5$|$64.2\pm10.4$|$61.6\pm11$ | $58.6\pm20.8$  | $34.7\pm11.8$  | $23.3\pm5.6$ |
> | pen-cloned |$28.5\pm6.7$ |$39.2$|$32.1\pm7.5$| **$49\pm9.5$**  |  $43.4\pm20.3$    |$20.8\pm7.3$  | $19.0\pm7.5$  |
> | Average      |$52.6$   | $38.3$    | $48.1$  |  **$55.3$**   | $51.0$  | $27.7$ |  $21.1$|
>
> Their hypepameters are: IQL $\tau=0.7, \lambda=2.0$ (tuned among $[0.6, 0.7, 0.8, 0.9] \times [1.0, 2.0, 3.0]$, The AWAC uses $\lambda=1.0$ (tuned among $[0.1, 0.2, 0.3, 0.5, 0.8, 1.0, 1.2]$) and Td3BC uses $\alpha=0.1$ (tuned among $[0.01, 0.1, 0.2, 0.5]$). AWAC-MCEP $\tilde{\lambda}=10.0, \lambda^E = 1000$, TD3BC-MCEP $\tilde{\alpha}=0.1, \alpha^E=0.1$.
>
> TD3BC under this high BC coefficiency is the only method that outperforms behavior cloning. However, the critic learning presents exponentially high values during the training of AWAC and TD3BC. In these cases, the MCEP fails to utilize the arbitrarily high values provided by the critics, resulting in worse performance.
>
> > So, MCEP can only aid policy constraint methods? Can it benefit IQL as well?
> >
> The MCEP is desgined overcome an overlooked drawback of policy constraints methods, hence it aims to aid policy constraints methods. This weakness is that most policy constraints methods do not distinguish 1) avoiding OOD actions to address unstable value learning and 2) obtaining inference-time performance. Instead, Policy constraints methods treat 1) and 2) as one problem and expect one identical policy constraint to solve both. This is the source of the wellknown tradeoff (Fujimoto et al 2018, Kumar et al 2019). The general idea behihd MCEP is to address 1) and 2) separately, hence a simple two-constraint framework is devised.
>
> IQL is also an instance that follows this general idea and implicitly has two policy constraints. Its value learning can be transformed to actor-critic with implicit \textit{target policy} (Hansen-Estruch et al., 2023). Its policy extraction derives an evaluation policy that potentially has a different constraint from the target policy.
>
> Pessimistic value methods such as CQL also have the same drawback by solving 1) and 2) with the same degree of conservative. How to address this drawback for pessimism methods is also an open problem (Hong et al 2022, Hong et al 2023).
>
> **References**
>
> Hansen-Estruch, P., Kostrikov, I., Janner, M., Kuba, J.G. and Levine, S., 2023. Idql: Implicit q-learning as an actor-critic method with diffusion policies. arXiv preprint arXiv:2304.10573.
>
> Hong, J., Kumar, A. and Levine, S., 2022. Confidence-conditioned value functions for offline reinforcement learning. arXiv preprint arXiv:2212.04607.
>
> Hong, Z.W., Kumar, A., Karnik, S., Bhandwaldar, A., Srivastava, A., Pajarinen, J., Laroche, R., Gupta, A. and Agrawal, P., 2023. Beyond Uniform Sampling: Offline Reinforcement Learning with Imbalanced Datasets. arXiv preprint arXiv:2310.04413.

---

> > ### Comment · Reviewer_VtJK · 2023-11-21
> >
> > Thanks for your rebuttal. It is good to see the numerical results. I have updated my score, as promised. I cannot support more since this method seems to be restricted in the category of offline policy constraint methods, and its performance on some challenging tasks is somewhat unsatisfying. I also hold my opinion that the results on wider datasets other than MuJoCo are necessary. It is okay that the proposed method does not have advantages over baselines. As I commented, if there is a failure on other datasets, explanations are expected to be included such that the readers can capture both the strengths and weaknesses of MCEP. I also strongly recommend the authors add the numerical results on Adroit/AntMaze datasets into the revision.
> >
> > Nevertheless, I tend to like this work and what the authors have done so far. The weaknesses side does not downgrade the contribution of this work. Consequently, I believe this paper deserves a score of 6.

---

> > > ### Author Response · Authors · 2023-11-22
> > >
> > > Thank you for your constructive feedback.
> > >
> > > > its performance on some challenging tasks is somewhat unsatisfying.
> > >
> > > To figure out whether the MCEP can work in complex domains and to better address this concern, we conduct a new experiment on a diverse set of 16 robotic manipulation tasks (Jorge et al 2022, Hussing et al 2023): 16 tasks $\times$ 2-version datasets for each task.
> > >
> > > - Average Success (%) Rate among 16 robotic manipulation tasks
> > > | Dataset      | BC       | CRR    | TD3BC           | IQL | TD3BC-MCEP |
> > > | ------------ | --------- | ---------- | --------- | --------- | --------- |
> > > | Medium |29.0 | 56.9| 45.9| 55.5  | **59.6**|
> > > | Medium Replay |1.2 |11.9|15.0| 25.0 |  **26.4** |
> > >
> > > The above results present that the MCEP (based on TD3BC) improves the performance significantly in these complex tasks. For more details about their hyperparameters and the success rate for each task, please refer to the general response on the top of this page.
> > >
> > > Please let us know whether this experiment solves your remaining concern.
> > >
> > > **References**
> > >
> > > Jorge A. et al., CompoSuite: A compositional reinforcement learning benchmark. In 1st Conference on Lifelong Learning Agents, 2022a.
> > >
> > > Hussing, et al., 2023. Robotic Manipulation Datasets for Offline Compositional Reinforcement Learning. arXiv preprint arXiv:2307.07091.

---

### Official Review · Reviewer_r8JF · 2023-11-02

**Soundness:** 2 fair
**Presentation:** 2 fair
**Contribution:** 2 fair
**Rating:** 5
**Confidence:** 4

**Summary:**

The paper introduces a new approach to address the problems in offline policy learning (e.g. extrapolation). The idea is to train an extra policy (called evaluation policy) based on the $Q$ function learned from the critic of a standard constrained actor-critic offline method. The idea is that using different constraint weights for the critic and evaluation policy should addresses the trade-off between evaluation performance and stable value estimate.

**Strengths:**

The paper is easy to follow and well written. As far as I know the idea is novel.

Experiments seem to be well conducted and results are fairly explained.

**Weaknesses:**

While the idea of the approach is interesting, I think the paper needs a bit more work. My main concern is that the contribution is limited and the results are not super clear and convincing to me.

- For example, given that $\pi_e$ is not involved in the optimization of $Q$, why are you training $\pi_e$ at each step and not only at the end? Training $\pi_e$ at the end will allow to do an analysis of the impact of the constraint.
- Have you tried to train a greedy policy starting from the recovered Q, similarly to what done in the grid experiment?
- Why haven't you tested other environments in D4RL, eg Antmaze-v0?
- Figures are not readable when printed out. The font is too small.


Typos:

acheive -> achieve

priority. i.e.

wrong latex commands in page 9

**Questions:**

See Weaknesses part.

---

> ### Author Response · Authors · 2023-11-14
> **A Response to reviewer r8JF**
>
> We appreciate reviewer **r8JF** for their review of our manuscript.
>
> For your concern of contribution is limited, **we respectfully argue that our contributions do not emphasize strong algorithmic performance, but are two-fold, as concluded below:**
>
> 1) We propose to distinguish the distributional shifts for Bellman estimate error reduction and the inference-time performance. Existing policy constraints methods (Kumar et al 2019, Nair et al 2020, Wang et al., 2023) focus on addressing the first type but wish to achieve the second type.
>
> 2) We propose to address these two types of distributional shifts separately and propose a general algorithm that learns two policies with different policy constraints. Our experiments on both D4RL locomotion and the challenging humanoid illustrate that the MCEP empowers conventional methods (TD3BC and AWAC) to achieve near SOTA performance and shows a consistent performance improvement on SOTA algorithm DQL.
>
> >For example, given that $\pi_e$ is not involved …
>
> We provide a new experiment in Appendix Section A.4 to illustrate the difference between these two design options. Our algorithm optimizes two policies together to allow parallelizing their learning thus the training time is the same as the base algorithms. This is powerful for algorithms that require long training time for the policy (e.g. DQL which utilizes diffusion policies).
>
> >Have you tried to train a greedy policy …
>
> While this idea works for the grid example, a greedy policy in larger MDPs will inquire the q function about OOD actions where the estimated Q is arbitrarily high. Based on this, the evaluation policy still needs a policy constraint (though milder). Pessimism Q methods (CQL etc) do not have this issue so their policy does not need the policy constraints.
>
> >Why have’nt you tested other environments …
>
> As this is asked by reviewers **r8JF, VtJK, WVL4 (3/4 of the reviewers),** we answer this question in the general review that can be found at the top of this page.
>
> >Figures are not readable when printed out.
>
> We apologize for the inconvenience during your review. We changed the font size in the new manuscript (as well as fixing the typos). Again, we appreciate your effort and suggestion.
>
> Please let us know whether our answers address all your concerns. We are more than happy to answer more if you have further questions and to take more suggestions to improve our manuscript.
>
> **References:**
>
> Kumar, A., Fu, J., Soh, M., Tucker, G. and Levine, S., 2019. Stabilizing off-policy q-learning via bootstrapping error reduction. *Advances in Neural Information Processing Systems*, *32*.
>
> Nair, A., Gupta, A., Dalal, M. and Levine, S., 2020. Awac: Accelerating online reinforcement learning with offline datasets. *arXiv preprint arXiv:2006.09359*.
>
> Wang, Z., Hunt, J.J. and Zhou, M., 2022. Diffusion policies as an expressive policy class for offline reinforcement learning. *arXiv preprint arXiv:2208.06193*.

---

> > ### Comment · Reviewer_r8JF · 2023-11-14
> > **Response**
> >
> > Thank you for the answers. I appreciate your clarifications, but I still have concerns about the novelty of the approach. For example, there are several methods that are not compared or tested in your work.
> >
> > Approximate policy improvement at test time is a well-studied topic. To my knowledge, Wang et al. introduced this idea in their paper "Critic Regularized Regression" (NeurIPS 2020). They also presented experiments on challenging environments, such as the humanoid.
> >
> > An even simpler and more common approach for policy improvement at test time is to sample actions and take the empirical argmax (or constrained argmax) at evaluation. I would like to see these approaches integrated into your paper.

---

> > > ### Author Response · Authors · 2023-11-16
> > >
> > > Thanks for your reply and thoughtful suggestion. We made the following changes to address your concern.
> > >
> > > - We are glad to have added the CRR (Wang et al., NeurIPS 2020) to our experiments (Section 5.3) on the humanoid tasks. We show that TD3BC-MCEP still outperforms all the baselines. More details about the implementation and hyperparameter tuning for CRR are described with highlighted text in Section 5.3 and Appendix A.2.
> > > - We conduct an investigation of these on-the-fly inference-time methods on humanoid tasks. Our experiments include both the Weighted Policy Constraint (WPC) from CRR and the Argmax. To make a fair comparison, we tune the values of these methods’ hyperparameters. The results in Appendix A.4 show that these methods brought performance improvement on both the TD3BC and TD3BC-MCEP. It enables TD3BC to be competitive with TD3Bc-MCEP on *medium* tasks but fails to do so on *medium-replay* and *medium-expert*.
> > >
> > > Please let us know whether all your concerns have been resolved.

---

> > > > ### Author Response · Authors · 2023-11-22
> > > >
> > > > > Why have’nt you tested other environments …
> > > >
> > > > To better address this concern, we conduct a new experiment on a diverse set of 16 robotic manipulation tasks(Jorge et al 2022, Hussing et al 2023). For each task, we select two datasets (medium and medium-replay). The following results present that the MCEP is able to provide significant performance improvement even in complex tasks. For more details about their hyperparameters and the success rate for each task, please refer to the general response on the top of this page.
> > > >
> > > >
> > > > - Average Success (%) Rate among 16 robotic manipulation tasks
> > > > | Dataset      | BC       | CRR    | TD3BC           | IQL | TD3BC-MCEP |
> > > > | ------------ | --------- | ---------- | --------- | --------- | --------- |
> > > > | Medium |29.0 | 56.9| 45.9| 55.5  | **59.6**|
> > > > | Medium Replay |1.2 |11.9|15.0| 25.0 |  **26.4** |
> > > >
> > > > We hope these new experiment results, as well as the experiments provided in our last reply, have addressed your concerns. We are pleased to take questions if anything unclear.
> > > >
> > > > **References**
> > > >
> > > > Jorge A. et al., CompoSuite: A compositional reinforcement learning benchmark. In 1st Conference on Lifelong Learning Agents, 2022a.
> > > >
> > > > Hussing, et al., 2023. Robotic Manipulation Datasets for Offline Compositional Reinforcement Learning. arXiv preprint arXiv:2307.07091.

---

> > > > > ### Comment · Reviewer_r8JF · 2023-12-05
> > > > > **Thank you for the rebuttal**
> > > > >
> > > > > I want to thank the authors for the clarifications and additional results provided in the rebuttal. The authors have clarified a few of my doubts. While I'm okay with increasing the score to 6, I still believe the paper stands in a borderline position.

---

### Author Response · Authors · 2023-11-14
**General Response**

# General Response

We want to thank the reviewers for their thorough reviews and insightful feedback. To address specific points raised by the reviewers, we reply to all reviewers in the individual threads.

## Summary of Positive Feedback

It appears reviewers have recognized various strengths in our work.

- **Motivation:** Reviewer VtJK recognizes “this paper is well-motivated”,  reviewer px9k notes “two kinds of distributional shifts … are rarely distinguished in offline RL literature”, and reviewer WVL4 writes “the toy example illustrates the motivation well” (3/4 of the reviewers).
- **Methods:** Reviewer VtJK mentions “An ingenious idea” and “quite simple and easy to implement”, px9k writes “simple and general”.
- **Experiment**, r8JF notes that “experiments seem to be well conducted”, VtJK comments “the improvements are significant on many base algorithms”, and px9k mentions “MCEP demonstrates good performance.” (3/4 of the reviewers).
- **“well written”** is commented by the reviewer r8JF, VtJK and WVL4 (3/4 of the reviewers)

## Experimental Design and Contribution

Reviewers **r8JF**, **VtJK**, and **WVL4** raised the question of not presenting the evaluation on environments such as AntMaze, Kitchen, and Adroit.

The **AntMaze** is a long-horizon and sparse-reward environment and the datasets lack of optimal trajectories. As a result, these tasks require a high-quality value estimate. Policy constraints methods (Kumar et al 2019, Nair et al 2020, Fujimoto et al 2021, Wang et al., 2023) do not provide stable learning in these environments. The SOTA method DQL reports good performance while its training is still unstable (Wang et al., 2023). The MCEP, as mentioned in the limitation section, may not recover its performance with unstable learning of the base algorithm. So it does not outperform the baselines in AntMaze environments.

The **Kitchen and Adroit** require strong behavior regularization.The policy constraint methods are able to achieve mediocre performance in the kitchen and adroit with a high behavior cloning coefficient (Tarasov et al 2023). At the same time, the critic shows exponentially high values. The MCEP can hardly utilize arbitrarily high Q values, therefore, it fails to outperform baselines.

**We argue that our contributions do not emphasize strong algorithmic performance, but are two-fold, as concluded below:**

1) We provide an insight into distributional shifts, by distinguishing it’s influence on Bellman estimate error and inference-time performance separately. Existing policy constraints methods (Kumar et al 2019, Nair et al 2020, Wang et al., 2023) mainly focus on addressing the Bellman estimate error but expect the policy to achieve good inference performance.

This contribution is supported by experiments:

[Section 5.1] Empirical analysis of the toy maze

[Sections 5.2 and A.2] Constraint turning on the locomotion tasks

2) We propose to address these two types of distributional shifts separately. The resulting method is a simple but general approach to achieve both the bellman error reduction and the inference-time performance. It empowers conventional policy constraints methods to achieve near SOTA performance and provide a consistent performance improvement on SOTA method DQL.

This contribution is supported by experiments:

[Section 5.3] Evaluation of D4RL locomotion tasks and high-dimensional Humanoid tasks

[Sections 5.4 and 5.5] An ablation study on the extra evaluation policy and its constraint strengths

**Reference:**

Kumar, A., Fu, J., Soh, M., Tucker, G. and Levine, S., 2019. Stabilizing off-policy q-learning via bootstrapping error reduction. *Advances in Neural Information Processing Systems*, *32*.

Nair, A., Gupta, A., Dalal, M. and Levine, S., 2020. Awac: Accelerating online reinforcement learning with offline datasets. *arXiv preprint arXiv:2006.09359*.

Fujimoto, S. and Gu, S.S., 2021. A minimalist approach to offline reinforcement learning. *Advances in neural information processing systems*, *34*, pp.20132-20145.

Wang, Z., Hunt, J.J. and Zhou, M., 2022. Diffusion policies as an expressive policy class for offline reinforcement learning. *arXiv preprint arXiv:2208.06193*.

Tarasov, D., Kurenkov, V., Nikulin, A. and Kolesnikov, S., 2023. Revisiting the Minimalist Approach to Offline Reinforcement Learning. *arXiv preprint arXiv:2305.09836*.

---

> ### Author Response · Authors · 2023-11-22
>
> To better resolve a common concern raised from reviewers, we conduct an experiment on a diverse set of 16 robotic manipulation tasks on the KUKA’s IIWA robot, from the CompoSuite (Jorge et al 2022, Hussing et al 2023). Below we show the average success rates among 100 episodes of evaluating the policy from the last step of a 1-million step training. Results are averaged among 5 random seeds.
>
> ## Averaged Success Rates for 16 Robotic Manipulation tasks
>
> ### Medium
>
> | Dataset      | BC       | CRR    | TD3BC           | IQL | TD3BC-MCEP |
> | ------------ | --------- | ---------- | --------- | --------- | --------- |
> | Box-PickPlace            |10.8   | 52.2    |89.8       | 93.8  | 100|
> | Box-Push                 |74.6   | 20.6     |93.8       | 91.8  |   99.8   |
> | Box-Shelf           |91.8   | 50.2        |93.2  |  98.6  | 99.2  |
> | Box-Trashcan        |8.6    | 62   |1.2  |  0  | 0  |
> | Dumbbell-PickPlace  |38.6   | 60.8    | 63.2  |  86.8   | 70.4  |
> | Dumbbell-Push       |55.2   | 12.6    | 54.0  |  66.6  | 58.0  |
> | Dumbbell-Shelf      |40.8   | 67    | 21.0  |  0.6   | 44.6  |
> | Dumbbell-Trashcan      |5.2 | 74    | 28.0  |  87.1   | 68.2  |
> | Hollowbox-PickPlace |42.4   | 92.8    | 82.6  |  95.2   | 92.2 |
> | Hollowbox-Push      |0      | 39.2    | 49.2  |  69.4   | 98.2  |
> | Hollowbox-Shelf      |72.2  | 91.6    | 95.4  |  98.2   | 98.4  |
> |Hollowbox-Trashcan      |0   | 24    | 0  |  0  | 0  |
> | Plate-PickPlace      |0     | 85.8    | 2.2  |  1   | 0.4  |
> | Plate-Push       |0.6       | 55.2    | 0 |  0   | 25.0 |
> | Plate-Shelf      |24.2      | 94    | 60.4  |  99   | 99.8  |
> | Plate-Trashcan   |0.2       | 29.2    | 0.8  |  0.4   | 0  |
> | **Average**               |29.0      | 56.9    | 45.9  |  55.5   | **59.6**  |
>
> ### Medium-Replay
> | Dataset      | BC       | CRR    | TD3BC           | IQL | TD3BC-MCEP |
> | ------------ | --------- | ---------- | --------- | --------- | --------- |
> | Box-PickPlace            |0   | 25.6|23.0          | 50.8  | 0|
> | Box-Push                 |0   |1.4 |60.8          | 0  |  41.6    |
> | Box-Shelf           |0   | 2.6    | 6.4  |  16.6  | 49.8  |
> | Box-Trashcan        |0   | 17    | 0  |  92.3   | 76.2  |
> | Dumbbell-PickPlace  |0   | 31.4    | 8.2  |  34.1   | 0  |
> | Dumbbell-Push       |4.1   | 0    | 24.2  |  3.2   | 55.8  |
> | Dumbbell-Shelf      |9.8   | 0.8    | 25.4  |  11.6   | 12.0  |
> | Dumbbell-Trashcan      |0   | 32.8    | 29.0  |  65.0   | 95.2  |
> | Hollowbox-PickPlace      |0   | 41.4    | 6.6  |  0.2   | 0.4  |
> | Hollowbox-Push      |0   | 2.8    | 23.0  |  30.0   | 3.0  |
> | Hollowbox-Shelf      |0   | 1.8    | 32.0  |  61.4   | 58.4  |
> |Hollowbox-Trashcan      |0   | 1.4    | 0  |  4.8  | 0  |
> | Plate-PickPlace      |0   | 7    | 2.2  |  29.4   | 0  |
> | Plate-Push       |0   | 18.2    | 0 |  0   | 10.6  |
> | Plate-Shelf      |0   | 2    | 0  |  0   | 19.4  |
> | Plate-Trashcan   |0.8   | 4.4    | 0.4  |  1.0   | 0  |
> | **Average**               |1.2   | 11.9    | 15.0  |  25.0   | **26.4**  |
>
>
> We consider similar hypermeter for all tasks. We tune CRR $\lambda = [0.4, 0.6, 0.8, 1.0, 1.2]$ and uses softmax inference-time policy improvement (see discussion with reviewer r8JF). For TD3BC $\alpha = [1.0, 2.0, 3.0, 4.0]$. IQL uses $\tau=0.7, \lambda=3.0$ as recommended in (Hussing et al). For TD3BC-MCEP, $\tilde{\alpha}=2, \alpha^E = [4.0, 6.0, 8.0, 10.0]$.
>
> In this experiment, we observe that the MCEP is able to provide consistent and significant performance improvement in complex tasks. These results should resolve the concern of whether the MCEP works in complex domains.
>
> **References**
>
> Jorge A. et al., CompoSuite: A compositional reinforcement learning benchmark. In 1st Conference on Lifelong Learning Agents, 2022a.
>
> Hussing, et al., 2023. Robotic Manipulation Datasets for Offline Compositional Reinforcement Learning. arXiv preprint arXiv:2307.07091.